# FALCON: An ML Framework for Fully Automated Layout-Constrained Analog Circuit Design

**Asal Mehradfar**[1]    **Xuzhe Zhao**[2]    **Yilun Huang**[2]    **Emir Ceyani**[1]
**Yankai Yang**[2]    **Shihao Han**[2]    **Hamidreza Aghasi**[2]    **Salman Avestimehr**[1]
[1]University of Southern California    [2]University of California, Irvine

mehradfa@usc.edu

## Abstract

Designing analog circuits from performance specifications is a complex, multi-stage process encompassing topology selection, parameter inference, and layout feasibility. We introduce FALCON, a unified machine learning framework that enables fully automated, specification-driven analog circuit synthesis through topology selection and layout-constrained optimization. Given a target performance, FALCON first selects an appropriate circuit topology using a performance-driven classifier guided by human design heuristics. Next, it employs a custom, edge-centric graph neural network trained to map circuit topology and parameters to performance, enabling gradient-based parameter inference through the learned forward model. This inference is guided by a differentiable layout cost, derived from analytical equations capturing parasitic and frequency-dependent effects, and constrained by design rules. We train and evaluate FALCON on a large-scale custom dataset of 1M analog mm-wave circuits, generated and simulated using Cadence Spectre across 20 expert-designed topologies. Through this evaluation, FALCON demonstrates >99% accuracy in topology inference, <10% relative error in performance prediction, and efficient layout-aware design that completes in under 1 second per instance. Together, these results position FALCON as a practical and extensible foundation model for end-to-end analog circuit design automation. Our code and dataset are publicly available at https://github.com/AsalMehradfar/FALCON.

## 1   Introduction

Analog radio frequency (RF) and millimeter-wave (mm-wave) circuits are essential to modern electronics, powering critical applications in signal processing [1], wireless communication [2], sensing [3], radar [4], and wireless power transfer systems [5]. Despite their importance, the design of analog circuits remains largely manual, iterative, and dependent on expert heuristics [6–8]. This inefficiency stems from several challenges: a vast and continuous design space that is difficult to explore systematically; tightly coupled performance metrics (e.g. gain, noise, bandwidth, and power) that create complex trade-offs; and physical and layout-dependent interactions that complicate design decisions. As demand grows for customized, high-performance analog blocks, this slow, expert-driven design cycle has become a critical bottleneck. While machine learning (ML) has revolutionized digital design automation, analog and RF circuits still lack scalable frameworks for automating the full pipeline from specification to layout.

While recent ML approaches have made progress in analog circuit design, they typically target isolated sub-tasks such as topology generation or component sizing [9, 10] at the schematic level, without addressing the full synthesis pipeline. Many efforts assume fixed topologies [11–14], limiting adaptability to new specifications or circuit families. Optimization strategies often rely on black-box methods that do not scale well to large, continuous design spaces [15]. Some methods predict

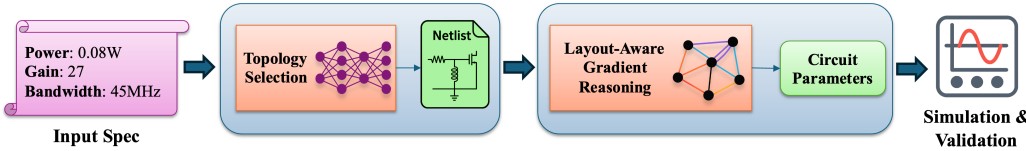

Figure 1: Our AI-based circuit design pipeline. Given a target performance specification, FALCON first selects a suitable topology, then generates design parameters through layout-aware gradient-based reasoning with GNN model. Then, the synthesized circuit is validated using Cadence simulations.

performance metrics directly from netlists [16], but do not support inverse design, i.e., generating circuit parameters from target specifications. Furthermore, layout awareness is typically handled as a separate post-processing step [17], missing the opportunity to guide optimization with layout constraints. Finally, many available benchmarks are built on symbolic or synthetic simulations [18], lacking the fidelity and realism of the process of commercial grade design flows. As a result, current ML pipelines do not allow fully generalizable, layout-aware, and end-to-end analog circuit design.

We propose FALCON (Fully Automated Layout-Constrained analOg circuit desigN), a scalable and modular machine learning framework for end-to-end analog and RF circuit design. Built on a dataset of over one million Cadence-simulated circuits, FALCON comprises three core components (Figure 1): (1) a lightweight multilayer perceptron (MLP) selects the most appropriate topology given a target performance specification; (2) a generalizable graph neural network (GNN) maps circuit topology and element-level parameters to performance metrics, operating on a native graph representation derived from Cadence netlists; and (3) gradient-based optimization over the forward GNN model recovers design parameters that meet the target specification, guided by a differentiable layout-aware loss that encodes parasitic effects and physical constraints. Notably, the GNN model in FALCON generalizes effectively to unseen topologies, enabling inverse design across diverse circuit families, even in low-data regimes, with optional fine-tuning for improved accuracy. By integrating layout modeling directly into the optimization process, FALCON unifies schematic and physical considerations within a single differentiable learning framework.

Our main contributions are as follows:

- We construct a **large-scale analog/RF circuit dataset** comprising over one million Cadence-simulated datapoints across 20 expert-designed topologies and five circuit types.
- We introduce a **native netlist-to-graph representation** that preserves both structural and parametric fidelity, enabling accurate learning over physical circuit topologies.
- We design a **generalizable GNN** capable of accurate performance prediction and parameter inference across both seen and unseen topologies, with optional fine-tuning.
- We develop a **modular ML framework** for end-to-end inverse design, incorporating performance-driven topology selection and layout-aware gradient-based optimization, with a differentiable loss that enforces area constraints, design-rule compliance, and frequency-dependent modeling of passive components.
- We show that FALCON enables **fast, reliable inverse design** under layout and physical constraints, generating high-quality circuits in under one second per instance on CPU.

## 2 Related Work

While recent ML-based approaches have advanced analog and RF circuit design, they typically target isolated stages of the design flow—such as topology generation, parameter sizing, or schematic-level performance prediction—without supporting unified, end-to-end synthesis. FALCON bridges this gap by jointly addressing aforementioned stages within a single framework.

**Topology generation** methods aim to select or synthesize candidate circuit structures [9, 19, 20], often using discrete optimization or generative models to explore the circuit graph space. However, these approaches typically target low-frequency or simplified designs [9] and may produce physically invalid or non-manufacturable topologies. In contrast, FALCON leverages a curated set of netlists, ensuring manufacturable validity and eliminating the need to rediscover fundamental circuit structures.

**Parameter sizing and performance prediction** have been explored through various learning paradigms. Reinforcement learning [10, 21] and Bayesian optimization [15, 22] optimize parameters via trial-and-error, often requiring large simulation budgets. Supervised learning methods [23, 24, 11] regress parameter values from performance targets under fixed topologies. Graph-based models [16] incorporate topology-aware representations to predict performance metrics from netlists. However, these approaches focus on forward prediction or black-box sizing and do not support inverse design across varied topologies. In contrast, FALCON unifies forward modeling and parameter inference in a single differentiable architecture that generalizes to unseen netlists.

**Layout-aware sizing and parasitic modeling** have been explored to mitigate schematic-to-layout mismatch. Parasitic-aware methods [25] integrate pre-trained parasitic estimators into Bayesian optimization loops for fixed schematics. While effective for estimation, these approaches rely on time-consuming black-box search and lack inverse design capabilities. Other methods, such as ALIGN [26] and LayoutCopilot [27], generate layouts from fully sized netlists using ML-based constraint extraction or scripted interactions, but assume fixed parameters and do not support co-optimization or differentiable inverse design. In contrast, FALCON embeds layout objectives directly into the learning loss, enabling joint optimization of sizing and layout without relying on external parasitic models. For mm-wave circuits, our layout-aware loss captures frequency-sensitive constraints via simplified models that implicitly reflect DRC rules, EM coupling, and performance-critical factors such as quality factor and self-resonance frequency.

**Datasets** for analog design are often limited to symbolic SPICE simulations or small-scale testbeds that do not reflect real-world design flows. AnalogGym [18] and AutoCkt [13] rely on synthetic circuits and symbolic simulators, lacking the process fidelity, noise characteristics, and layout-dependent behavior of foundry-calibrated flows. In contrast, FALCON is trained on a large-scale dataset constructed from over one million Cadence-simulated circuits across 20 topologies and five circuit categories, offering a substantially more realistic foundation for ML-driven analog design.

To the best of our knowledge, FALCON is the first framework to unify topology selection, parameter inference, and layout-aware optimization in a single end-to-end pipeline, validated at scale using industrial-grade Cadence simulations for mm-wave analog circuits. A qualitative comparison with representative prior work is provided in Appendix A.

## 3 A Large-Scale Dataset and Inverse Design Problem Formulation

### 3.1 Dataset Overview

We construct a large-scale dataset of analog and RF circuits simulated using industry-grade Cadence tools [28] with a 45nm CMOS process design kit (PDK). The dataset spans five widely used mm-wave circuit types for wireless applications [29, 30]: low-noise amplifiers (LNAs) [31–34], mixers [35–38], power amplifiers (PAs) [39–43], voltage amplifiers (VAs) [44–48], and voltage-controlled oscillators (VCOs) [49–53]. Each circuit type is instantiated in four distinct topologies, resulting in a total of 20 expert-designed architectures.

For each topology, expert-designed schematics were implemented in Cadence Virtuoso, and key design parameters were manually identified based on their functional relevance. Parameter ranges were specified by domain experts and systematically swept using Cadence ADE XL, enabling parallelized Spectre simulations across the design space. For each configuration, performance metrics—such as gain, bandwidth, and oscillation frequency—were extracted and recorded. Each datapoint therefore includes the full parameter vector, the corresponding Cadence netlist, and the simulated performance metrics. The resulting dataset comprises over one million datapoints, capturing a wide range of circuit behaviors and design trade-offs across diverse topologies. This large-scale, high-fidelity dataset forms the foundation for training and evaluating our inverse design pipeline. Detailed metric definitions and per-topology parameter ranges appear in Appendix B.

### 3.2 Graph-Based Circuit Representation

To enable flexible and topology-agnostic learning, we represent each analog circuit as a graph extracted from its corresponding Cadence netlist. Nodes correspond to voltage nets (i.e., electrical connection points), and edges represent circuit elements such as transistors, resistors, capacitors, or sources. Multi-terminal devices—such as transistors and baluns—are decomposed into multiple

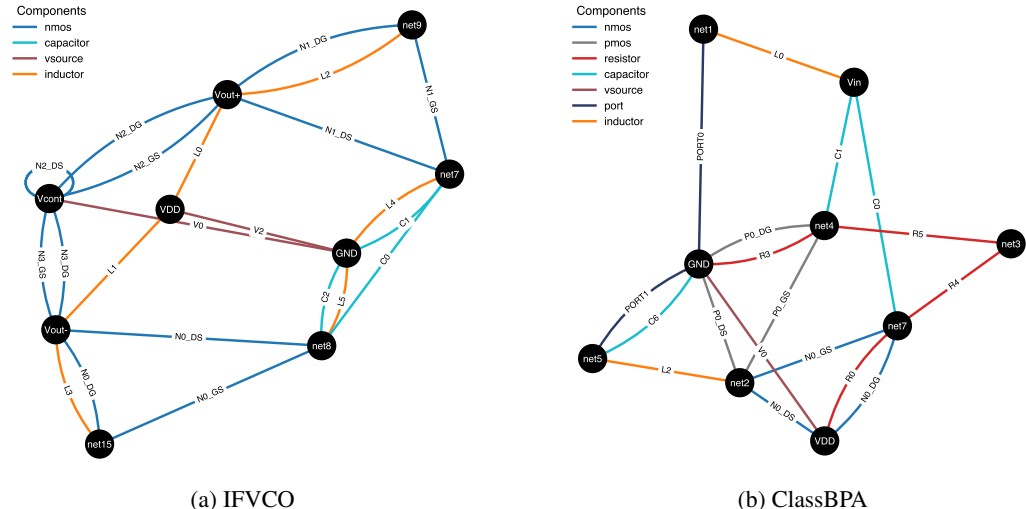

(a) IFVCO
(b) ClassBPA

Figure 2: Graph representations of two analog circuit topologies from our dataset: (a) IFVCO and (b) ClassBPA. Nodes represent electrical nets, and colored edges denote circuit components such as transistors, capacitors, inductors, and sources. Each component type is visually distinguished by color and labeled with its name and terminal role (e.g., N2_GS, V0). For transistors, labels such as GS, DS, and DG denote source-to-gate, drain-to-source, and drain-to-gate connections, respectively. These graphs serve as input to our GNN-based performance modeling and inverse design pipeline.

edges, and multiple components may connect the same node pair, resulting in **heterogeneous, multi-edged graphs** that preserve structural and functional diversity.

Recent works such as DICE [54] have explored transistor-level circuit-to-graph conversions for self-supervised learning, highlighting the challenges of faithfully capturing device structure and connectivity. In contrast, our approach maintains a native representation aligned with foundry-compatible netlists. Rather than flattening or reinterpreting device abstractions, we preserve symbolic parameters, multi-edge connections, and device-specific edge decomposition directly from the schematic source, enabling scalable learning across diverse analog circuit families.

To support learning over such structured graphs, each edge is annotated with a rich set of attributes: (i) a **categorical device type**, specifying the component and connected terminal pair (e.g., NMOS drain–gate, resistor); (ii) **numeric attributes**, such as channel length or port resistance, fixed by the schematic; (iii) **parametric attributes**, defined symbolically in the netlist (e.g., *W1*, *R3*) and resolved numerically during preprocessing; (iv) **one-hot categorical features**, such as source type (DC, AC, or none); and (v) **computational attributes**, such as diffusion areas (*Ad*, *As*) derived from sizing. This rule-based graph construction generalizes across circuit families without task-specific customization. Graphs in the FALCON dataset range from 4–40 nodes and 7–70 edges, reflecting the variability of practical analog designs. Figure 2 shows two representative graph examples from our dataset—IFVCO and ClassBPA.

### 3.3 Inverse Design Problem Definition

In analog and RF circuit design, the traditional modeling process involves selecting a topology $T$ and parameter vector $x$, then evaluating circuit behavior via simulation to obtain performance metrics $y = f(T, x)$. This forward workflow depends heavily on designer intuition, manual tuning, and exhaustive parameter sweeps. Engineers typically simulate many candidate $(T, x)$ pairs and select the one that best satisfies the target specification—a slow, costly, and unguided process.

In contrast, our goal is to perform *inverse design*: given a target performance specification $y_{\text{target}}$, we aim to directly infer a topology and parameter configuration $(T, x)$ such that $f(T, x) \approx y_{\text{target}}$, without enumerating the full design space. This inverse problem is ill-posed and the search space is constrained by both device-level rules and layout-aware objectives.

Formally, the task is to find the optimal topology $T^* \in \mathcal{T}$ and the optimal parameters $x^* \in \mathbb{R}^p$ such that $f(T^*, x^*) \approx y_{\text{target}}$ where $f : \mathcal{T} \times \mathbb{R}^p \to \mathbb{R}^d$ the true performance function implemented by expensive Cadence simulations. In practice, $f$ is nonlinear and non-invertible, making direct inversion intractable. FALCON addresses this challenge through a modular, three-stage pipeline:

**Stage 1: Topology Selection.** We frame topology selection as a classification problem over a curated set of $K$ candidate topologies $\{T_1, \ldots, T_K\}$. Given a target specification $y_{\text{target}}$, a lightweight MLP selects the topology $T^* \in \mathcal{T}$ most likely to satisfy it, reducing the need for exhaustive search.

**Stage 2: Performance Prediction.** Given a topology $T$ and parameter vector $x$, we train a GNN $f_\theta$ to predict the corresponding performance $\hat{y} = f_\theta(T, x)$. This model emulates the forward behavior of the simulator $f$, learning a continuous approximation of circuit performance across both seen and unseen topologies. By capturing the topology-conditioned mapping from parameters to performance, $f_\theta$ serves as a differentiable surrogate that enables gradient-based inference in the next stage.

**Stage 3: Layout-Aware Gradient Reasoning.** Given $y_{\text{target}}$ and a selected topology $T^*$, we infer a parameter vector $x^*$ by minimizing a loss over the learned forward model $f_\theta$. Specifically, we solve:

$$x^* = \arg\min_x \ \mathcal{L}_{\text{perf}}(f_\theta(T^*, x), y_{\text{target}}) + \lambda \, \mathcal{L}_{\text{layout}}(x), \tag{1}$$

where $\mathcal{L}_{\text{perf}}$ measures prediction error, and $\mathcal{L}_{\text{layout}}$ encodes differentiable layout-related constraints such as estimated area and soft design-rule penalties. Optimization is performed via gradient descent, allowing layout constraints to guide the search through a physically realistic parameter space.

## 4 Stage 1: Performance-Driven Topology Selection

**Task Setup.** We formulate topology selection as a supervised classification task over a fixed library of 20 expert-designed circuit topologies $\mathcal{T} = \{T_1, T_2, \ldots, T_{20}\}$. Rather than generating netlists from scratch—which often leads to invalid or impractical circuits—we select from a vetted set of designer-verified topologies. This ensures that all candidates are functionally correct, layout-feasible, and manufacturable. While expanding the topology set requires retraining, our lightweight MLP classifier enables rapid updates, making the approach scalable. This formulation also aligns with practical design workflows, where quickly identifying a viable initial topology is critical.

Each datapoint is represented by a 16-dimensional performance vector of key analog/RF metrics[1]. We normalize features using z-scores computed from the training set. Missing metrics (e.g., oscillation frequency for amplifiers) are imputed with zeros, yielding zero-centered, fixed-length vectors that retain task-relevant variation. Dataset splits are stratified to preserve class balance across training, validation, and test sets. We assume each target vector is realizable by at least one topology in $\mathcal{T}$, though the library can be extended with new designs.

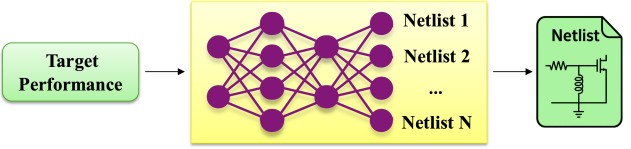

Figure 3: In Stage 1, an MLP classifier selects the most suitable circuit topology from a library of human-designed netlists, conditioned on the target performance specification.

Table 1: Classification performance on topology selection.

| Metric | Score (%) |
|---|---|
| Accuracy | 99.57 |
| Balanced Accuracy | 99.33 |
| Macro Precision | 99.27 |
| Macro Recall | 99.33 |
| Macro F1 | 99.30 |
| Micro F1 | 99.57 |

**Model Architecture and Training.** We train a 5-layer MLP with hidden size 256 and ReLU activations for this problem. The model takes the normalized performance vector $y_{\text{target}} \in \mathbb{R}^{16}$ as input and outputs a probability distribution over 20 candidate topologies. The predicted topology is selected as $T^* = \arg\max_{T_k \in \mathcal{T}} \text{MLP}(y_{\text{target}})_k$. We train the model using a cross-entropy loss and the Adam optimizer [55], with a batch size of 256. An overview of this process is shown in Figure 3.

---

[1]DC power consumption (DCP), voltage gain (VGain), power gain (PGain), conversion gain (CGain), $S_{11}$, $S_{22}$, noise figure (NF), bandwidth (BW), oscillation frequency (OscF), tuning range (TR), output power (OutP), $P_{\text{SAT}}$, drain efficiency (DE), power-added efficiency (PAE), phase noise (PN), voltage swing (VSwg).

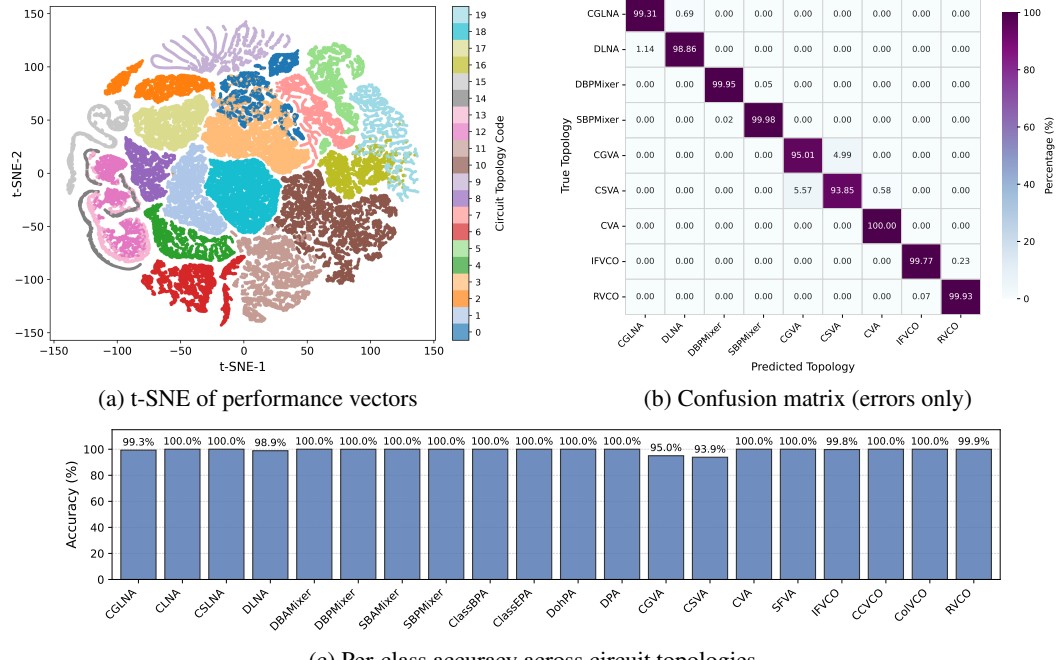

(a) t-SNE of performance vectors       (b) Confusion matrix (errors only)

(c) Per-class accuracy across circuit topologies

Figure 4: Topology selection results. (a) Performance vectors form well-separated clusters in t-SNE space, showing that circuit functionality is semantically predictive of topology. (b) Misclassifications primarily occur among voltage amplifier variants with overlapping gain-bandwidth tradeoffs. (c) Per-class test accuracy exceeds 93% across all 20 circuit topologies.[2]

**Evaluation.** We begin by assessing the quality of the input representation used for topology classification. Normalized performance vectors encode rich semantic information about circuit behavior. To validate this, we project them into a two-dimensional t-SNE space [56] (Figure 4(a)). The resulting clusters align closely with topology labels, indicating that performance specifications reflect underlying schematic structure and are effective inputs for supervised classification.

We assess classification performance using accuracy, balanced accuracy, macro precision, macro recall, macro F1, and micro F1 scores on the test set. As summarized in Table 1, the classifier achieves an overall accuracy of 99.57%, with macro F1 of 99.30% and balanced accuracy of 99.33%, demonstrating strong generalization across all 20 circuit topologies. Micro F1 (identical to accuracy in the multiclass setting) reaches 99.57%, while macro metrics—averaged equally across classes—highlight robustness to class imbalance. Seed-averaged results with 95% confidence intervals are provided in Appendix C. These trends are reinforced by the per-class accuracy plot in Figure 4(c), where most topologies reach 100% accuracy.

The confusion matrix in Figure 4(b) visualizes only the misclassified instances, as most classes achieve perfect accuracy. The few observed errors are primarily concentrated among the two voltage amplifier topologies—common-gate (CGVA) and common-source (CSVA). These circuits operate near the gain-bandwidth limit of the transistor, and when the main amplifier transistor size is held constant, performance metrics such as power consumption, gain, and bandwidth can converge across these architectures. This occasional overlap in the performance space introduces ambiguity in classification for a small subset of instances. For other circuit categories, no significant confusion is expected or observed. These results validate our hypothesis that performance vectors contain sufficient semantic structure for accurate, scalable topology classification.

---

[2]The 20 circuit topologies—listed in the same order as the numerical labels in Figure 4(a)—are: CGLNA (Common Gate), CLNA (Cascode), CSLNA (Common Source), DLNA (Differential), DBAMixer (Double-Balanced Active), DBPMixer (Double-Balanced Passive), SBAMixer (Single-Balanced Active), SBPMixer (Single-Balanced Passive), ClassBPA (Class-B), ClassEPA (Class-E), DohPA (Doherty), DPA (Differential), CGVA (Common Gate), CSVA (Common Source), CVA (Cascode), SFVA (Source Follower), IFVCO (Inductive-Feedback), CCVCO (Cross-Coupled), ColVCO (Colpitts), RVCO (Ring).

# 5 Stage 2: Generalizable Forward Modeling for Performance Prediction

**Task Setup.** The goal of Stage 2 is to learn a differentiable approximation of the circuit simulator that maps a topology $T$ and parameter vector $x$ to a performance prediction $\hat{y} = f_\theta(T, x)$, where $\hat{y} \in \mathbb{R}^{16}$. Unlike black-box simulators, this learned forward model enables efficient performance estimation and supports gradient-based parameter inference in Stage 3. The model is trained to generalize across circuit families and can be reused on unseen topologies with minimal fine-tuning.

Each datapoint consists of a graph-structured Cadence netlist annotated with resolved parameter values and the corresponding performance metrics. We frame the learning task as a supervised regression problem. Since not all performance metrics apply to every topology (e.g., oscillation frequency is undefined for amplifiers), we train the model using a **masked mean squared error** loss:

$$\mathcal{L}_{\text{masked}} = \frac{1}{\sum_i m_i} \sum_{i=1}^{d} m_i \cdot (\hat{y}_i - y_i)^2, \tag{2}$$

where $m_i = 1$ if the $i$-th metric is defined for the current sample, and $0$ otherwise.

**Model Architecture and Training.** Each circuit is represented as an **undirected multi-edge graph** with voltage nets as nodes and circuit components as edges. All circuit parameters—both fixed and sweepable—are assigned to edges, along with categorical device types and one-hot encoded indicators. For each edge $e$, these attributes are concatenated to form a unified feature vector $x_e$. The feature set is consistent within each component type but varies across types (e.g., NMOS vs. inductor), reflecting the structure defined in Section 3.2.

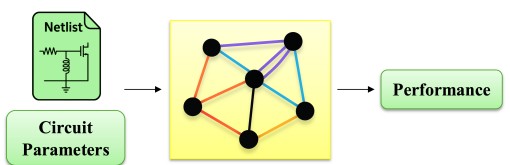

Figure 5: In Stage 2, a custom edge-centric GNN maps an undirected multi-edge graph constructed from the circuit netlist to a performance vector.

To account for component heterogeneity, we apply **type-specific MLP encoders** $\phi_{\text{enc}}^{(t_e)}$ to the raw features $x_e$ of each edge $e$, producing edge embeddings $z_e = \phi_{\text{enc}}^{(t_e)}(x_e)$, where $t_e$ denotes the component type of edge $e$. Node features are initialized as scalar dummy values and mapped to a hidden dimension via a shared linear layer. We then apply a 4-layer **edge-centric message-passing GNN** with residual connections. At each layer $\ell$, every node $u$ receives messages from all incident edges $e \in \mathcal{E}_u$, where $\mathcal{E}_u$ denotes the set of edges connected to $u$. Messages are computed using a shared function $\phi_{\text{MSG}}$, which takes the hidden state of the neighboring node $\text{src}(e)$ and the associated edge embedding. Node states are then updated via a learnable function $\phi_{\text{UPD}}$, followed by a residual connection and nonlinearity:

$$m_u^{(\ell)} = \sum_{e \in \mathcal{E}_u} \phi_{\text{MSG}}\left(h_{\text{src}(e)}^{(\ell)}, z_e\right), \quad h_u^{(\ell+1)} = \text{ReLU}\left(\phi_{\text{UPD}}\left(m_u^{(\ell)}\right) + h_u^{(\ell)}\right)$$

where $z_e \in \mathbb{R}^d$ is the learned embedding of edge $e$, and $h_{\text{src}(e)}^{(\ell)} \in \mathbb{R}^d$ is the hidden state of the node at the other end of edge $e$ from $u$ at layer $\ell$. The update function $\phi_{\text{UPD}}$ is a shared linear transformation applied to all nodes. After $L = 4$ GNN layers, node embeddings are aggregated using **global mean pooling** to obtain a fixed-size graph representation:

$$z_{\text{graph}} = \frac{1}{|V|} \sum_{u \in V} h_u^{(L)},$$

where $V$ is the set of all nodes in the graph. The resulting vector is then passed through a fully connected MLP (`output_mlp`) to predict the 16-dimensional performance vector $\hat{y} \in \mathbb{R}^{16}$. An overview of this GNN-based forward prediction pipeline is shown in Figure 5.

To stabilize training, physical parameters are rescaled by their expected units (e.g. resistance by $10^3$), and performance targets are normalized to z-scores using training statistics. We train the model using the Adam optimizer (learning rate $10^{-3}$, batch size 256) and a `ReduceLROnPlateau` scheduler. Xavier uniform initialization is used for all layers, and early stopping is based on validation loss. We adopt the same splits as in Section 4 for consistency in evaluation.

**Evaluation.** We evaluate the accuracy of the GNN forward model $f_\theta$ on a test set drawn from 19 of the 20 topologies. One topology—RVCO—is entirely excluded from training, validation, and test splits to assess generalization to unseen architectures. Prediction quality is measured using standard regression metrics: coefficient of determination ($R^2$), root mean squared error (RMSE), and mean absolute error (MAE), computed independently for each of the 16 performance metrics. We also report the *mean relative error per metric*, computed as the average across all test samples where each metric is defined. As summarized in Table 2, the model achieves high accuracy across all dimensions, with an average $R^2$ of 0.972.

Table 2: Prediction accuracy of the forward GNN on all 16 circuit performance metrics.

| Metric | DCP | VGain | PGain | CGain | $S_{11}$ | $S_{22}$ | NF | BW | OscF | TR | OutP | $P_{SAT}$ | DE | PAE | PN | VSwg |
|---|---|---|---|---|---|---|---|---|---|---|---|---|---|---|---|---|
| Unit | mW | dB | dB | dB | dB | dB | dB | GHz | GHz | GHz | dBm | dBm | % | % | dBc/Hz | mV |
| R² | 1.0 | 1.0 | 0.99 | 1.0 | 0.93 | 1.0 | 0.99 | 0.98 | 0.97 | 0.83 | 0.97 | 1.0 | 1.0 | 1.0 | 0.89 | 1.0 |
| RMSE | 0.27 | 0.101 | 0.536 | 0.833 | 1.515 | 0.21 | 0.534 | 0.972 | 0.723 | 0.293 | 0.91 | 0.095 | 0.226 | 0.143 | 2.536 | 0.071 |
| MAE | 0.198 | 0.072 | 0.208 | 0.188 | 0.554 | 0.12 | 0.2 | 0.376 | 0.184 | 0.097 | 0.238 | 0.066 | 0.163 | 0.105 | 1.159 | 0.046 |
| Rel. Err. | 11.2% | 2.6% | 19.0% | 6.1% | 11.4% | 1.9% | 4.5% | 6.5% | 0.6% | 6.5% | 4.6% | 4.4% | 4.6% | 11.0% | 1.3% | 1.4% |

To evaluate end-to-end prediction accuracy at the sample level, we compute the *mean relative error per instance*, defined as the average relative error across all valid (non-masked) performance metrics for each test sample. Figure 6 shows the distribution of this quantity across the test set (trimmed at the 95th percentile to reduce the impact of outliers). The distribution is sharply concentrated, indicating that most predictions closely match their corresponding target vectors. Without percentile trimming, the overall mean relative error across the full test set is **9.09%**. Seed-averaged results with 95% confidence intervals are provided in Appendix C.

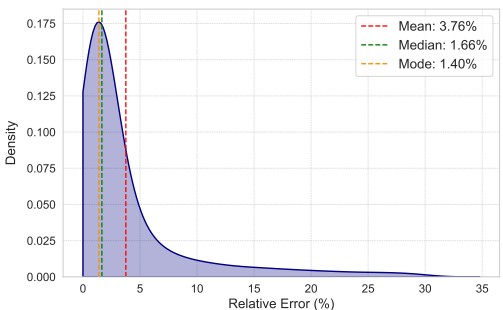

Figure 6: Distribution of relative error (%) across the test set for the GNN forward model. Plot is trimmed at the 95th percentile.

**Generalizing to Unseen Topologies via Fine-Tuning.** To assess the generalization ability of our pretrained GNN, we evaluate it on the held-out RVCO topology, which was entirely excluded from the Stage 2 training, validation, and test splits. Notably, the RVCO training partition used here matches that of the Stage 1 experiments (Section 4), enabling consistent cross-stage evaluation.

We fine-tune the GNN by freezing all encoder and message-passing layers and updating only the final output head (`output_mlp`). Fine-tuning is performed on the RVCO training set, which contains approximately 30,000 instances, and completes in under 30 minutes on a MacBook CPU.

Even in the zero-shot setting—where the model has never seen RVCO topologies—the pretrained GNN achieves a nontrivial mean relative error of 30.4%, highlighting its strong cross-topology generalization. Fine-tuning reduces this error to just **0.9%**, demonstrating that the structural and parametric priors learned during pretraining are highly transferable. Table 3 reports detailed performance across five key metrics of RVCO, confirming that the pretrained GNN can be rapidly adapted to novel circuit families with minimal supervision.

Table 3: Fine-tuning results on the held-out RVCO topology. Only the output head is updated using RVCO samples.

| Metric | DCP | OscF | TR | OutP | PN |
|---|---|---|---|---|---|
| Unit | W | GHz | GHz | dBm | dBc/Hz |
| R² | 1.0 | 1.0 | 1.0 | 0.97 | 0.98 |
| RMSE | 0.643 | 0.324 | 0.026 | 0.099 | 0.953 |
| MAE | 0.508 | 0.256 | 0.02 | 0.077 | 0.619 |
| Rel. Err. | 0.75% | 0.85% | 1.63% | 0.69% | 0.73% |

## 6 Stage 3: Layout-Aware Parameter Inference via Gradient Reasoning

**Task Setup.** Given a target performance vector $y_{\text{target}}$ and a selected topology $T^*$, the goal of Stage 3 is to recover a parameter vector $x^*$ that minimizes a total loss combining performance error and layout-aware penalties, using the learned forward model $f_\theta$ from Stage 2. This formulation enables instance-wise inverse design without requiring circuit-level simulation.

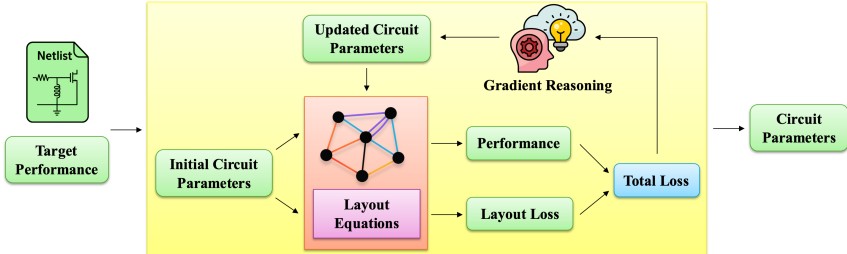

Figure 7: In Stage 3, gradient reasoning iteratively updates parameters to minimize a loss combining performance error and layout cost, computed via a differentiable analytical model.

To initialize optimization, we perturb domain-specific scale factors (e.g., $10^{-12}$ for capacitors) to sample a plausible starting point $x_0$. Parameters are iteratively updated via gradient descent, guided by both functional and physical objectives. Topology-specific constants are held fixed, and parameter values are clipped to remain within valid domain bounds throughout the process.

**Loss Function.** The total loss follows the structure defined in Eqn 1, jointly minimizing performance mismatch and layout cost:

$$\mathcal{L}_{\text{total}} = \mathcal{L}_{\text{perf}} + \lambda_{\text{area}} \cdot \mathcal{L}_{\text{layout}} \cdot g(\mathcal{L}_{\text{perf}}), \tag{3}$$

where $\mathcal{L}_{\text{perf}}$ is the masked mean squared error (see Eqn 2) between predicted and target performance vectors, and $\mathcal{L}_{\text{layout}}$ is a normalized area penalty derived from analytical layout equations. To prioritize functionality, layout loss is softly gated by a sigmoid function:

$$g(\mathcal{L}_{\text{perf}}) = 1 - \sigma\left(\gamma(\mathcal{L}_{\text{perf}} - \tau)\right),$$

where $\sigma(\cdot)$ denotes the sigmoid function, and $\gamma, \tau$ are fixed scalars controlling the sharpness and center of the gating. This gating attenuates layout penalties when performance error exceeds a threshold $\tau$, encouraging the model to first achieve functionality before optimizing for layout compactness.

We set $\tau = 0.05$, $\gamma = 50$, and normalize layout area by $1\,\text{mm}^2$ to stabilize gradients. The layout weight $\lambda_{\text{area}} = 0.02$ is chosen empirically to balance performance accuracy and physical realism without dominating the loss. This gated formulation supports manufacturable parameter recovery and reflects the broader paradigm of physics-informed learning [57]. Further discussion on user-defined objectives is provided in Appendix D.

**Differentiable Layout Modeling.** In mm-wave analog design, layout is not a downstream concern but a critical determinant of circuit performance—particularly for passive components. Substrate coupling, proximity effects, and DRC-imposed geometries directly affect key metrics such as resonance frequency, quality factor, and impedance matching. To incorporate these effects, we introduce a differentiable layout model that computes total physical area analytically from circuit parameters. This enables layout constraints to directly guide parameter optimization during inverse design. By minimizing the layout area in distributed mm-wave circuits [58], unwanted signal loss [59] is reduced, the self-resonance frequency of passives can increase [60], and phase and amplitude mismatches across signal paths [61] can be reduced.

The layout model is deterministic and non-learned. It estimates area contributions from passive components—capacitors, inductors, and resistors—as these dominate total area and exhibit layout-sensitive behavior. Active devices (e.g., MOSFETs) are excluded since their geometries are fixed by the PDK and are negligible [62]. For a given parameter vector $x$, the total layout loss is computed as: $\mathcal{L}_{\text{layout}}(x) = \sum_{e \in \mathcal{E}_{\text{passive}}} A_e(x)$, where $\mathcal{E}_{\text{passive}}$ is the set of passive elements, and $A_e(x)$ is the area of the created layout for the passive component based on analytical physics-based equations. The area of element $e$ is estimated based on its 2D dimensions (e.g., $A = W \cdot L$ for resistors and capacitors). This area is normalized and used as a differentiable penalty in the optimization objective (see Eqn 3). Further implementation details are provided in Appendix E.

**Gradient Reasoning Procedure.** Starting from the initialized parameter vector $x_0$, we iteratively update parameters via gradient reasoning. At each step, the frozen forward model $f_\theta$ predicts the performance $\hat{y} = f_\theta(T, x)$, and the total loss $\mathcal{L}_{\text{total}}$ is evaluated. Gradients are backpropagated with respect to $x$, and updates are applied using the Adam optimizer. Optimization proceeds for a fixed number of steps, with early stopping triggered if the loss fails to improve over a predefined window.

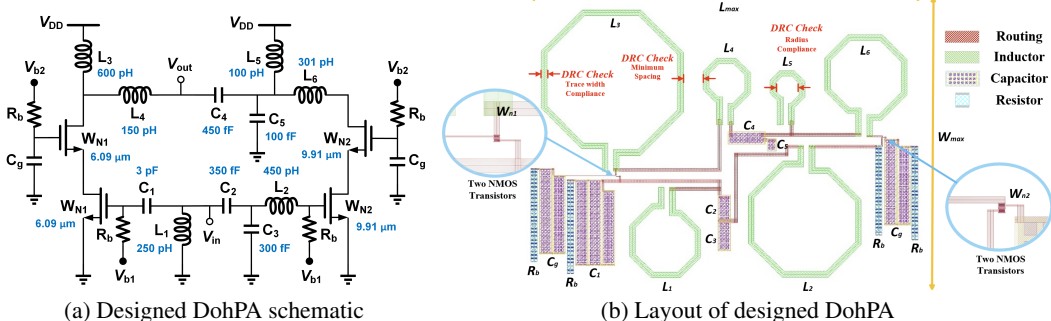

(a) Designed DohPA schematic

(b) Layout of designed DohPA

Figure 8: Stage 3 results for a synthesized DohPA. The schematic (a) reflects optimized parameters to meet the target specification. The layout (b) is DRC-compliant and physically realizable. The final design achieves a mean relative error of 5.4% compared to the target performance.

To handle varying circuit difficulty and initialization quality, we employ an adaptive learning rate strategy. Each instance begins with a moderate learning rate ($10^{-6}$), refined during optimization via a `ReduceLROnPlateau` scheduler. If the solution fails to meet thresholds on performance error or layout area, optimization restarts with a more exploratory learning rate. This adjustment balances exploration and fine-tuning, enabling rapid convergence to physically valid solutions, typically within milliseconds to under one second per instance. An overview is shown in Figure 7.

**Evaluation.** We evaluate Stage 3 on 9,500 test instances (500 per topology) using our gradient-based optimization pipeline. A design is considered *converged* if it meets both: (i) a predicted mean relative error below 10%, and (ii) a layout area under a topology-specific bound—1 mm$^2$ for most circuits and 1.5 mm$^2$ for DLNA, DohPA, and ClassBPA. The 10% error threshold reflects the forward model's $\sim 9\%$ average prediction error (Section 5). A design is deemed *successful* if its final Cadence-simulated performance deviates from the target by less than 20%, confirming real-world viability. Our method achieves a success rate of **78.5%** and a mean relative error of **17.7%** across converged designs, with average inference time **under 1 second** on a MacBook CPU. Notably, success rate is coupled with the convergence threshold: tighter error bounds yield higher accuracy but require more iterations—critical for large-scale design tasks.

To illustrate the effectiveness of our pipeline, Figure 8 shows a representative result for the DohPA topology: the synthesized schematic is shown on the left, and the corresponding layout is on the right. These results confirm that the recovered parameters are both functionally accurate and physically realizable. Together, they demonstrate that FALCON enables layout-aware inverse design within a single differentiable pipeline—a capability not supported by existing analog design frameworks.

## 7 Conclusion and Future Work

We presented FALCON, a modular framework for end-to-end analog and RF circuit design that unifies topology selection, performance prediction, and layout-aware parameter optimization. Trained on over one million Cadence-simulated mm-wave circuits, FALCON combines a lightweight MLP, a generalizable GNN, and differentiable gradient reasoning to synthesize circuits from specification to layout-constrained parameters. FALCON achieves >99% topology selection accuracy, <10% prediction error, and efficient inverse design—all within sub-second inference. In addition, the GNN forward model generalizes to unseen topologies with minimal fine-tuning, supporting broad practical deployment. Further discussion of training and inference efficiency, as well as practical limitations, is provided in Appendix F.

In future work, we aim to expand the topology library and support hierarchical macroblocks for scalable design beyond the cell level. We also plan to extend the dataset to cover multiple operating frequencies, enabling validation across diverse bands, and to enhance the layout-aware optimization with learned parasitic models, EM-informed constraints, and electromigration considerations for more accurate post-layout estimation. Finally, integrating reinforcement learning or diffusion-based models for generative topology synthesis represents a promising step toward general-purpose analog design automation.

## Acknowledgments and Disclosure of Funding

We thank Andrea Villasenor and Tanqin He for their assistance with circuit data generation. We also thank Mohammad Shahab Sepehri for his insightful discussions and thoughtful feedback during the development of this work.

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

# A  Qualitative Comparison with Prior Work

To contextualize FALCON within the broader landscape of analog circuit design automation, we provide a qualitative comparison against representative prior works in Table 4. This comparison spans key capabilities including topology selection, parameter inference, performance prediction, layout awareness, and simulator fidelity. We additionally assess reproducibility via dataset and code availability, and introduce a new axis—**RF/mm-wave** support—to highlight methods evaluated on high-frequency circuit blocks such as LNAs, mixers, and VCOs. Compared to existing approaches, FALCON is the only framework that unifies all these dimensions while maintaining foundry-grade fidelity and open-source accessibility. Definitions for each comparison axis are provided in Table 5.

Table 4: Qualitative comparison of FALCON with prior works across key capabilities in analog circuit design automation.

| Method | Topology Selection | Parameter Inference | Performance Prediction | Layout Awareness | Foundry Grade | RF/ mm-Wave | Public Dataset | Public Code |
|---|---|---|---|---|---|---|---|---|
| CktGNN [9] | ✔ | ✔ | ✗ | ✗ | ✗ (SPICE) | ✗ | ✔ | ✔ |
| LaMAGIC [19] | ✔ | ✗ | ✗ | ✗ | ✗ (SPICE) | ✗ | ✗ | ✗ |
| AnalogCoder [20] | ✔ | ✗ | ✗ | ✗ | ✗ (SPICE) | ✗ | ✔ | ✔ |
| GCN-RL [10] | ✗ | ✔ | ✗ | ✗ | ✔ (SPICE/Cadence) | ✗ | ✗ | ✗ (incomplete) |
| Cao et al. [21] | ✗ | ✔ | ✗ | ✗ | ✔ (ADS/Cadence) | ✗ | ✗ | ✗ |
| BO-SPGP [15] | ✗ | ✔ | ✔ | ✗ | ✔ (Cadence) | ✗ | ✗ | ✗ |
| ESSAB [22] | ✗ | ✔ | ✔ | ✗ | ✔ (Cadence) | ✗ | ✗ | ✗ |
| AICircuit [23, 24] | ✗ | ✔ | ✗ | ✗ | ✔ (Cadence) | ✔ | ✔ | ✔ |
| Krylov et al. [11] | ✗ | ✔ | ✗ | ✗ | ✗ (SPICE) | ✗ | ✔ | ✔ |
| Deep-GEN [16] | ✗ | ✗ | ✔ | ✗ | ✗ (SPICE) | ✗ | ✔ | ✔ |
| Liu et al. [25] | ✗ | ✗ | ✗ | ✔ | ✗ (SPICE + Parasitic Model) | ✔ | ✗ | ✗ |
| ALIGN [26] | ✗ | ✗ | ✗ | ✔ | ✔ (Cadence) | ✔ | ✔ | ✔ |
| LayoutCopilot [27] | ✗ | ✗ | ✗ | ✔ | ✔ (Cadence) | ✗ | ✗ | ✗ |
| AnalogGym [18] | ✗ | ✔ | ✗ | ✗ | ✗ (SPICE) | ✗ | ✔ | ✔ |
| AutoCkt [13] | ✗ | ✔ | ✗ | ✗ | ✔ (Cadence) | ✗ | ✗ | ✗ (incomplete) |
| L2DC [12] | ✗ | ✔ | ✗ | ✗ | ✗ (SPICE) | ✗ | ✗ | ✗ |
| CAN-RL [14] | ✗ | ✔ | ✗ | ✔ | ✔ (Cadence) | ✗ | ✗ | ✗ |
| AnGeL. [17] | ✔ | ✔ | ✔ | ✗ | ✗ (SPICE) | ✗ | ✗ | ✗ |
| **FALCON (This work)** | ✔ | ✔ | ✔ | ✔ | ✔ (Cadence) | ✔ | ✔ | ✔ |

Table 5: Definitions of each comparison axis in Table 4.

| Column | Definition |
|---|---|
| **Topology Selection** | Does the method automatically select or predict circuit topology given a target specification? |
| **Parameter Inference** | Does the method infer element-level parameters (e.g., transistor sizes, component values) as part of design generation? |
| **Performance Prediction** | Can the method predict circuit performance metrics (e.g., gain, bandwidth, noise) from topology and parameters? |
| **Layout Awareness** | Is layout considered during optimization or training (e.g., via area constraints, parasitics, or layout-informed loss)? |
| **Dataset Fidelity** | Does the dataset reflect realistic circuit behavior (e.g., SPICE/Cadence simulations, PDK models)? |
| **RF/mm-Wave** | Is the method evaluated on at least one RF or mm-wave circuit type that reflects high-frequency design challenges? |
| **Public Dataset** | Is the dataset used in the work publicly released for reproducibility and benchmarking? |
| **Public Code** | Is the implementation code publicly available and documented for reproducibility? |

# B  Dataset Details and Performance Metric Definitions

During dataset generation, each simulated circuit instance is annotated with a set of performance metrics that capture its functional characteristics. All simulations are performed at a fixed frequency of 30 GHz, ensuring consistency across circuit types and relevance to mm-wave design. A total of 16 metrics are defined across all circuits—spanning gain, efficiency, impedance matching, noise, and frequency-domain behavior—though the specific metrics used vary by topology. For example, phase noise is only applicable to oscillators. An overview of all performance metrics is provided in Table 6.

## B.1  Low-Noise Amplifiers (LNAs)

Low-noise amplifiers (LNAs) are critical components in receiver front-ends, responsible for amplifying weak antenna signals while introducing minimal additional noise. Their performance directly influences downstream blocks such as mixers and analog-to-digital converters (ADCs), ultimately determining system-level fidelity [31]. To capture the architectural diversity of practical radio-frequency (RF) designs, we include four widely used LNA topologies in this study—common-source LNA (CSLNA), common-gate LNA (CGLNA), cascode LNA (CLNA), and differential LNA (DLNA)—as shown in Figure 9.

Table 6: Overview of 16 performance metrics used during dataset generation.

| Performance Name | Description |
|---|---|
| **DC Power Consumption (DCP)** | Total power drawn from the DC supply indicating energy consumption of the circuit |
| **Voltage Gain (VGain)** | Ratio of output voltage amplitude to input voltage amplitude |
| **Power Gain (PGain)** | Ratio of output power to input power |
| **Conversion Gain (CGain)** | Ratio of output power at the desired frequency to input power at the original frequency |
| **$S_{11}$** | Input reflection coefficient indicating impedance matching at the input terminal |
| **$S_{22}$** | Output reflection coefficient indicating impedance matching at the output terminal |
| **Noise Figure (NF)** | Ratio of input signal-to-noise ratio to output signal-to-noise ratio |
| **Bandwidth (BW)** | Frequency span over which the circuit maintains specified performance characteristics |
| **Oscillation Frequency (OscF)** | Steady-state frequency at which the oscillator generates a periodic signal |
| **Tuning Range (TR)** | Range of achievable oscillation frequencies through variation of control voltages |
| **Output Power (OutP)** | Power delivered to the load |
| **$P_{SAT}$** | Maximum output power level beyond which gain compression begins to occur |
| **Drain Efficiency (DE)** | Ratio of RF output power to DC power consumption. |
| **Power-Added Efficiency (PAE)** | Ratio of the difference between output power and input power to DC power consumption |
| **Phase Noise (PN)** | Measure of oscillator stability represented in the frequency domain at a specified offset |
| **Voltage Swing (VSwg)** | Maximum peak voltage level achievable at the output node |

The CSLNA is valued for its simplicity and favorable gain–noise trade-off, especially when paired with inductive source degeneration [30]. The CGLNA, often used in ultra-wideband systems, enables broadband input matching but typically suffers from a higher noise figure [32]. The CLNA improves gain–bandwidth product and reverse isolation, making it ideal for high-frequency, high-linearity applications [33]. The DLNA exploits circuit symmetry to enhance linearity and reject common-mode noise, and is commonly found in high-performance RF front-end designs [34]. The design parameters and performance metrics associated with these topologies are summarized in Table 7.

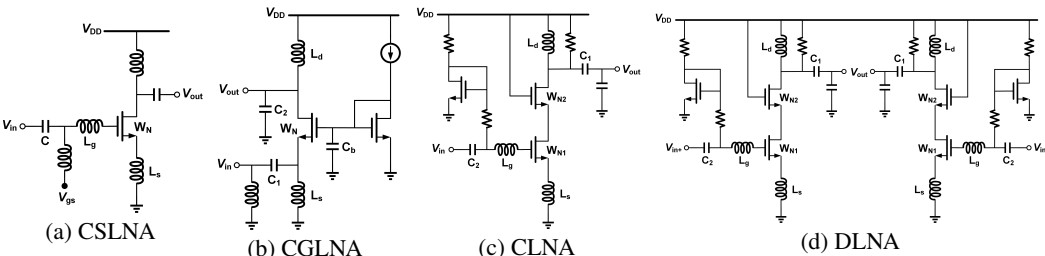

|   |   |   |   |
|---|---|---|---|
| (a) CSLNA | (b) CGLNA | (c) CLNA | (d) DLNA |

Figure 9: Schematic diagrams of the four LNA topologies.

Table 7: LNA topologies with parameter sweep ranges, sample sizes, and performance metrics.

| Dataset Type | Topology (Code) | # of Samples | Parameter | Sweep Range | Performance Metrics (Unit) |
|---|---|---|---|---|---|
| LNA | CGLNA (0) | 52k | $C_1$
$C_2$
$C_b$
$L_d$
$L_s$
$W_N$ | [100–600] fF
[50–300] fF
[250–750] fF
[80–580] pH
[0.5–5.5] nH
[12–23] μm | DCP (W)

PGain (dB)

$S_{11}$ (dB)

NF (dB)

BW (Hz) |
|  | CLNA (1) | 62k | $C_1, C_2$
$L_d$
$L_g$
$L_s$
$W_{N1}$
$W_{N2}$ | [50–250] fF
[140–300] pH
[0.4–2] nH
[50–250] pH
[3–5] μm
[7–9] μm |  |
|  | CSLNA (2) | 39k | $C$
$L_g$
$L_s$
$W_N$
$V_{gs}$ | [100–300] fF
[4–6] nH
[100–200] pH
[2.5–4] μm
[0.5–0.9] V |  |
|  | DLNA (3) | 92k | $C_1$
$C_2$
$L_d$
$L_g$
$L_s$
$W_{N1}$
$W_{N2}$ | [100–190] fF
[130–220] fF
[100–250] pH
[600–900] pH
[50–80] pH
[4–9.4] μm
[5–14] μm |  |

## B.2 Mixers

Mixers are fundamental nonlinear components in RF systems, responsible for frequency translation by combining two input signals to produce outputs at the sum and difference of their frequencies. This functionality is essential for transferring signals across frequency domains and is widely used in both transmission and reception paths [35]. To capture diverse mixer architectures, we implement four representative topologies in this work—double-balanced active mixer (DBAMixer), double-balanced passive mixer (DBPMixer), single-balanced active mixer (SBAMixer), and single-balanced passive mixer (SBPMixer)—as shown in Figure 10.

The DBAMixer integrates amplification and differential switching to achieve conversion gain and high port-to-port isolation. Despite its elevated power consumption and design complexity, it is well suited for systems requiring robust performance over varying conditions [36]. The DBPMixer features a fully differential structure that suppresses signal leakage and improves isolation, at the cost of signal loss and a strong local oscillator drive requirement [37]. The SBAMixer includes an amplification stage preceding the switching core to enhance signal strength and reduce noise, offering a balanced performance trade-off with increased power consumption and limited spurious rejection [30]. The SBPMixer employs a minimalist switching structure to perform frequency translation without active gain, enabling low power operation in applications with relaxed performance demands [38]. The parameters and performance metrics for these mixer topologies are listed in Table 8.

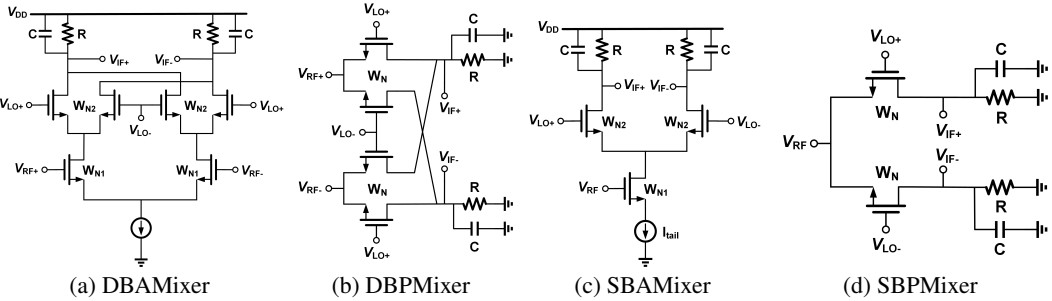

| (a) DBAMixer | (b) DBPMixer | (c) SBAMixer | (d) SBPMixer |

Figure 10: Schematic diagrams of the four Mixer topologies.

Table 8: Mixer topologies with parameter sweep ranges, sample sizes, and performance metrics.

| Dataset Type | Topology (Code) | # of Samples | Parameter | Sweep Range | Performance Metrics (Unit) |
|---|---|---|---|---|---|
| Mixer | DBAMixer (4) | 42k | C
R
$W_{N1}$
$W_{N2}$ | [1–10] pF
[1–10] kΩ
[10–30] μm
[5–25] μm | DCP (W)

CGain (dB)

NF (dB)

VSwg (V) |
| | DBPMixer (5) | 42k | C
R
$W_N$ | [100–500] fF
[100–600] Ω
[10–30] μm | |
| | SBAMixer (6) | 52k | C
R
$W_{N1}$
$W_{N2}$
$I_{tail}$ | [1–15] pF
[0.7–2.1] kΩ
[10–30] μm
[10–20] μm
[3–10] mA | |
| | SBPMixer (7) | 44k | C
R
$W_N$ | [1–30] pF
[1–30] kΩ
[5–29.5] μm | |

## B.3 Power Amplifiers (PAs)

Power amplifiers (PAs) are the most power-intensive components in radio-frequency (RF) systems and serve as the final interface between transceiver electronics and the antenna. Given their widespread use and the stringent demands of modern communication standards, PA design requires careful trade-offs across key performance metrics [39]. Based on the transistor operating mode, PAs are typically grouped into several canonical classes [40]. In this work, we implement four representative topologies—Class-B PA (ClassBPA), Class-E PA (ClassEPA), Doherty PA (DohPA), and differential PA (DPA)—as shown in Figure 11.

The ClassBPA employs complementary transistors to deliver high gain with moderate efficiency, making it suitable for linear amplification scenarios [41]. The ClassEPA uses a single transistor configured as a switch, paired with a matching network. By minimizing the overlap between drain voltage and current, this topology enables high-efficiency operation and improved robustness to component variation [30]. The DohPA combines main and peaking amplifiers using symmetric two-stack transistors, maintaining consistent gain and efficiency under varying power levels [42]. The DPA features a two-stage cascode structure designed to maximize gain and linearity, offering a favorable trade-off between output power and power consumption [43]. For this topology, we replace the transformer with a T-equivalent network to simplify modeling and training of the graph neural network. Parameter sweeps and performance metrics for these PAs are listed in Table 9.

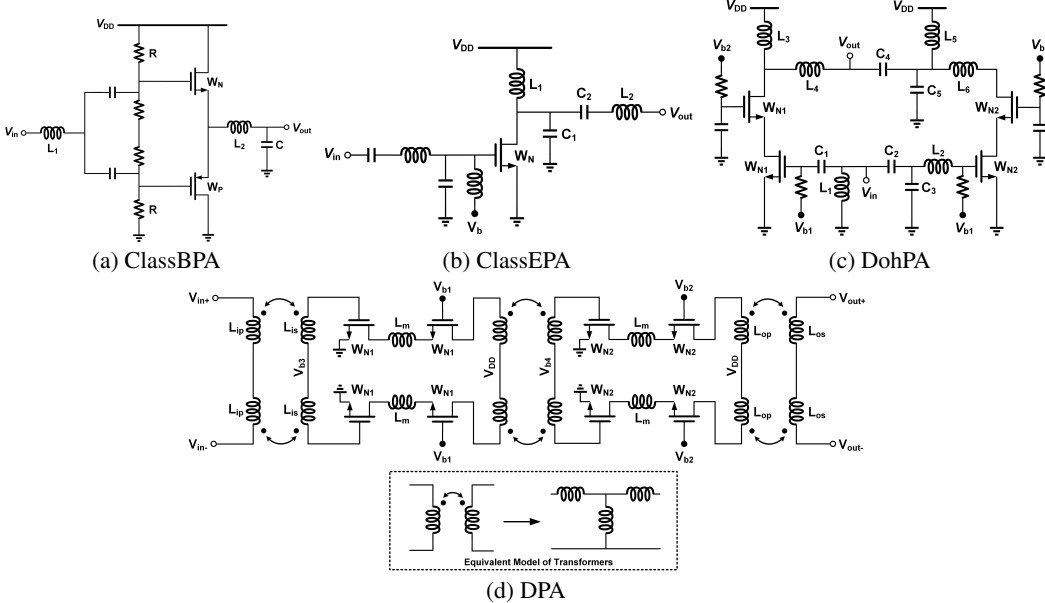

Figure 11: Schematic diagrams of the four PA topologies.

Table 9: PA topologies with parameter sweep ranges, sample sizes, and performance metrics.

| Dataset Type | Topology (Code) | # of Samples | Parameter | Sweep Range | Performance Metrics (Unit) |
|---|---|---|---|---|---|
| PA | ClassBPA (8) | 35k | C
$L_1$
$L_2$
R
$W_N$
$W_P$ | [55–205] fF
[1–1.4] nH
[1–8.5] pH
[1.5–4] kΩ
[10–20] μm
[3–8] μm | DCP (W)

PGain (dB)

$S_{11}$ (dB)

$S_{22}$ (dB)

$P_{SAT}$ (dBm)

DE (%)

PAE (%) |
| | ClassEPA (9) | 46k | $C_1$
$C_2$
$L_1$
$L_2$
$W_N$ | [100–200] fF
[500–700] fF
[100–300] pH
[100–150] pH
[15–30] μm | |
| | DohPA (10) | 120k | $C_1$
$C_2$
$C_3, C_5$
$C_4$
$L_1, L_5$
$L_2$
$L_3$
$L_4$
$L_6$
$W_{N1}, W_{N2}$ | [2–3] pF
[200–300] fF
[100–200] fF
[300–400] fF
[100–200] pH
[350–450] pH
[500–600] pH
[150–250] pH
[300–400] pH
[6–13] μm | |
| | DPA (11) | 80k | $L_{ip}$
$L_{is}$
$L_{op}$
$L_{os}$
$L_m$
$W_{N1}$
$W_{N2}$ | [100–500] pH
[300–700] pH
[0.8–1.2] nH
[400–800] pH
[50–250] pH
[6–31] μm
[10–35] μm | |

## B.4 Voltage Amplifiers (VAs)

Voltage amplifiers (VAs) are fundamental components in analog circuit design, responsible for increasing signal amplitude while preserving waveform integrity. Effective VA design requires balancing key performance metrics tailored to both RF and baseband operating conditions [44]. In this work, we implement four widely used VA topologies—common-source VA (CSVA), common-gate VA (CGVA), cascode VA (CVA), and source follower VA (SFVA)—as shown in Figure 12.

The CSVA remains the most widely adopted configuration due to its structural simplicity and high voltage gain. It is frequently used as the first gain stage in various analog systems [45]. The CGVA is suitable for applications requiring low input impedance and wide bandwidth, such as impedance transformation or broadband input matching [46]. The CVA, which cascades a common-source stage with a common-gate transistor, improves the gain–bandwidth product and enhances stability, making it ideal for applications demanding wide dynamic range and robust gain control [47]. The SFVA, also known as a common-drain amplifier, provides near-unity voltage gain and low output impedance, making it well suited for interstage buffering, load driving, and impedance bridging [48]. Parameter ranges and performance specifications for these VA topologies are listed in Table 10.

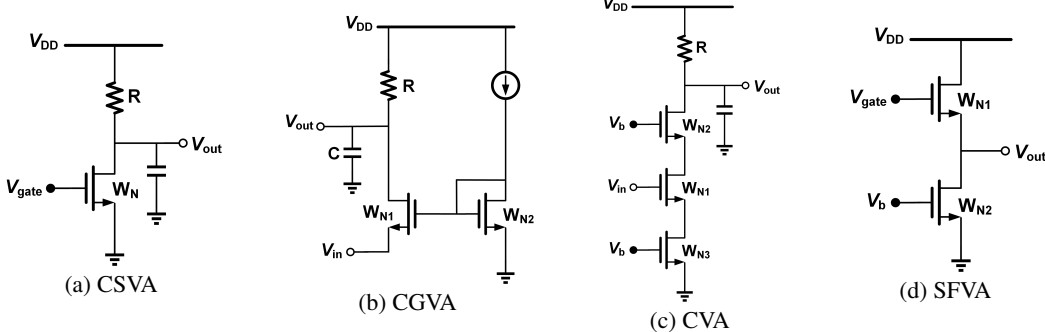

Figure 12: Schematic diagrams of the four VA topologies.

Table 10: VA topologies with parameter sweep ranges, sample sizes, and performance metrics.

| Dataset Type | Topology (Code) | # of Samples | Parameter | Sweep Range | Performance Metrics (Unit) |
|---|---|---|---|---|---|
| VA | CGVA (12) | 33k | C
R
$W_{N1}$
$W_{N2}$ | [0.1–1.5] pF
[0.1–1.5] kΩ
[5–30] μm
[5–10] μm | DCP (W)

VGain (dB)

BW (Hz) |
| | CSVA (13) | 21k | R
$W_N$
$V_{DD}$
$V_{gate}$ | [0.7–1.5] kΩ
[3–15] μm
[1–1.8] V
[0.6–0.9] V | |
| | CVA (14) | 22k | R
$W_{N1}, W_{N2}$
$W_{N3}$ | [1–3] kΩ
[1–10] μm
[10–15] μm | |
| | SFVA (15) | 28k | $W_{N1}$
$W_{N2}$
$V_{DD}$
$V_{gate}$
$V_b$ | [40–60] μm
[2–8] μm
[1.1–1.8] V
[0.6–1.2] V
[0.5–0.9] V | |

## B.5 Voltage-Controlled Oscillators (VCOs)

Voltage-controlled oscillators (VCOs) are essential building blocks in analog and RF systems, responsible for generating periodic waveforms with frequencies modulated by a control voltage. These circuits rely on amplification, feedback, and resonance to sustain stable oscillations. Owing to their wide tuning range, low power consumption, and ease of integration, VCOs are broadly used in systems such as phase-locked loops (PLLs), frequency synthesizers, and clock recovery circuits [49]. In this work, we implement four representative VCO topologies—inductive-feedback VCO (IFVCO), cross-coupled VCO (CCVCO), Colpitts VCO (ColVCO), and ring VCO (RVCO)—as shown in Figure 13.

The IFVCO employs an NMOS differential pair with an inductor-based feedback path to sustain oscillations. This topology provides favorable noise performance and compact layout, making it well suited for low-voltage, low-power designs [50]. The CCVCO achieves negative resistance through cross-coupling, enabling low phase noise and high integration density, and is widely adopted in frequency synthesizers and PLLs [51]. The ColVCO uses an LC tank and capacitive feedback to achieve high frequency stability and low phase noise, making it ideal for precision RF communication and instrumentation [52]. The RVCO consists of cascaded delay stages forming a feedback loop, offering low power consumption, wide tuning range, and minimal area footprint, though at the cost of higher phase noise. It is commonly used in on-chip clock generation and low-power sensor applications [53]. Design parameters and performance metrics for these VCO topologies are presented in Table 11.

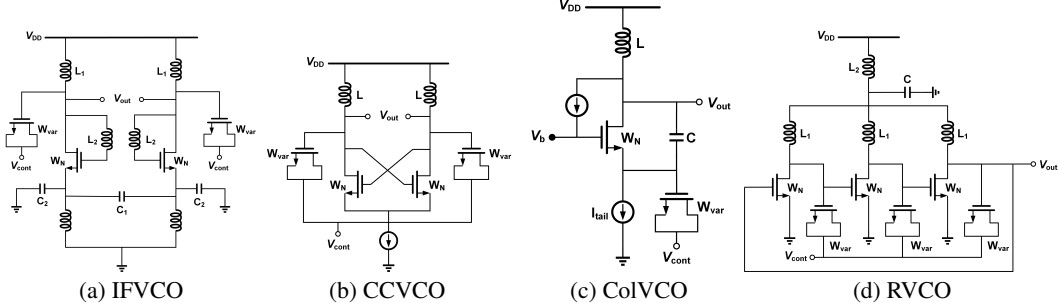

(a) IFVCO      (b) CCVCO      (c) ColVCO      (d) RVCO

Figure 13: Schematic diagrams of the four VCO topologies.

Table 11: VCO topologies with parameter sweep ranges, sample sizes, and performance metrics.

| Dataset Type | Topology (Code) | # of Samples | Parameter | Sweep Range | Performance Metrics (Unit) |
|---|---|---|---|---|---|
| VCO | IFVCO (16) | 43k | $C_1$
$C_2$
$L_1$
$L_2$
$W_N, W_{var}$ | [700–900] fF
[50–250] fF
[400–600] pH
[500–700] pH
[5–9] μm | DCP (W)

OscF (Hz)

TR (Hz)

OutP (dBm)

PN (dBc/Hz) |
| | CCVCO (17) | 54k | $L$
$W_N$
$W_{var}$ | [200–400] pH
[10–35] μm
[5–30] μm | |
| | ColVCO (18) | 90k | $C$
$L$
$W_N$
$W_{var}$
$V_b$
$I_{tail}$ | [80–140] fF
[250–350] pH
[30–50] μm
[5–15] μm
[0.7–1.2] V
[5–15] mA | |
| | RVCO (19) | 46k | $C$
$L_1$
$L_2$
$W_N$
$W_{var}$ | [300–700] fF
[300–500] pH
[50–250] pH
[20–40] μm
[5–25] μm | |

# C   Robustness Across Random Seeds

To evaluate the robustness of our models to random initialization and data shuffling, we repeated experiments using five distinct random seeds: {42, 123, 777, 2023, 3407}. Reporting across multiple seeds is important for ensuring that observed results are not artifacts of a specific initialization or training trajectory, but rather reflect the stable behavior of the method. For each metric, we compute the mean and 95% confidence interval across seeds, reporting results in the form $\mu \pm \Delta$.

Table 12: Topology selection performance with mean scores and 95% confidence intervals across five random seeds.

| Metric | Mean ± 95% CI (%) |
|---|---|
| Accuracy | 99.57 ± 0.01 |
| Balanced Accuracy | 99.34 ± 0.02 |
| Macro Precision | 99.27 ± 0.01 |
| Macro Recall | 99.34 ± 0.02 |
| Macro F1 | 99.30 ± 0.01 |
| Micro F1 | 99.57 ± 0.01 |

For the MLP topology selection model, results are highly stable across random seeds. The accuracy reaches $99.57 \pm 0.01\%$ with balanced accuracy at $99.34 \pm 0.02\%$, while both macro and micro F1 scores exceed $99.3\%$ with confidence intervals no larger than $\pm 0.02$. These narrow intervals indicate that the MLP's performance is effectively invariant to random initialization, underscoring its robustness and reliability in the topology selection stage of the pipeline (Section 4).

For the GNN-based forward performance prediction model, the overall mean relative error across all metrics is $9.14 \pm 0.38\%$ (95% CI). Individual performance predictions, including DC power consumption, gain, bandwidth, and oscillation frequency, exhibit narrow confidence intervals—for example, noise figure achieves $4.48 \pm 0.07\%$ error and oscillation frequency $0.65 \pm 0.03\%$. These results indicate that the GNN achieves consistently accurate predictions across diverse circuit characteristics. The tight confidence intervals further demonstrate that the model's performance is robust to random initialization, underscoring its reliability as a generalizable forward predictor within the pipeline (Section 5). The full seed-dependent results for both models are provided in Tables 12 and 13.

Table 13: Prediction accuracy of the forward GNN with mean scores and 95% confidence intervals across five random seeds.

| Metric | DCP | VGain | PGain | CGain | $S_{11}$ | $S_{22}$ | NF | BW | OscF | TR | OutP | $P_{SAT}$ | DE | PAE | PN | VSwg |
|--------|-----|-------|-------|-------|----------|----------|-----|-----|------|-----|------|-----------|-----|-----|-----|------|
| Rel. Err. | 11.64 | 3.10 | 18.46 | 5.25 | 11.49 | 1.94 | 4.48 | 6.28 | 0.65 | 6.55 | 4.86 | 4.31 | 4.51 | 11.58 | 1.34 | 1.71 |
| $\pm$ 95% CI (%) | $\pm 1.06$ | $\pm 0.42$ | $\pm 0.36$ | $\pm 0.44$ | $\pm 0.09$ | $\pm 0.15$ | $\pm 0.07$ | $\pm 0.43$ | $\pm 0.03$ | $\pm 0.04$ | $\pm 0.59$ | $\pm 0.24$ | $\pm 0.14$ | $\pm 1.72$ | $\pm 0.02$ | $\pm 0.29$ |

# D  User-Defined Loss Functions for Gradient Reasoning

Stage 3 of FALCON employs gradient reasoning with the forward GNN fixed, enabling the optimization objective to be redefined without retraining or fine-tuning the predictive model. This design allows users to flexibly adapt the loss function to capture specific trade-offs or constraints. We illustrate this flexibility with two examples.

**Weighted Performance Loss.** Rather than treating all performance metrics equally, users can specify weights $\alpha_i$ for each target metric:

$$\mathcal{L}_{\text{perf-weighted}} = \frac{1}{\sum_i m_i \alpha_i} \sum_{i=1}^{d} m_i \alpha_i \, (\hat{y}_i - y_i^{\text{target}})^2,$$

where larger $\alpha_i$ prioritize certain specifications (e.g., gain or noise figure). Here, $m_i = 1$ if the $i$-th metric is defined for the current sample, and 0 otherwise.

**Interval-Constrained Performance Loss.** Users may also define acceptable ranges for metrics rather than fixed targets. Given optional lower and/or upper bounds $y_i^{\text{lower}}, y_i^{\text{upper}}$, the interval penalty is:

$$\mathcal{L}_{\text{perf-interval}} = \frac{1}{\sum_i m_i} \sum_{i=1}^{d} m_i \left[ \mathbb{1}_{\{y_i^{\text{upper}} \text{ defined}\}} \max(0, \hat{y}_i - y_i^{\text{upper}}) + \mathbb{1}_{\{y_i^{\text{lower}} \text{ defined}\}} \max(0, y_i^{\text{lower}} - \hat{y}_i) \right],$$

where the indicator $\mathbb{1}_{\{\cdot\}}$ indicates whether the corresponding bound is specified. This formulation naturally handles the cases where only an upper bound, only a lower bound, or both bounds are provided. As above, $m_i = 1$ if the $i$-th metric is defined for the current sample, and 0 otherwise.

**General Extensibility.** More generally, the total loss in Eqn. 3 can be replaced with any user-defined formulation, allowing both $\mathcal{L}_{\text{perf}}$ and $\mathcal{L}_{\text{layout}}$ to be substituted with customized objectives. Additional physical constraints, multi-objective trade-offs, or alternative layout penalties can be incorporated with only a few lines of code. This extensibility underscores the flexibility of FALCON and enables the framework to adapt to diverse design objectives.

# E  Layout Design and DRC Compliance

## E.1  Design Rule Enforcement in 45 nm CMOS

We implemented FALCON using a 45 nm CMOS technology node, applying rigorous Design Rule Checking (DRC) at both the *cell* and *full-chip layout* levels. At the cell level, our parameterized layout generators enforced foundry-specific constraints, including minimum feature width and

length, contact and via spacing, and metal enclosure rules. At the circuit level, we incorporated physical verification to mitigate interconnect coupling, IR drop, and layout-dependent parasitic mismatches—factors that are especially critical in high-frequency and precision analog design.

DRC plays a vital role in ensuring that layouts comply with process design rules defined by the semiconductor foundry. Adhering to these rules ensures not only *physical manufacturability* but also *electrical reliability*. Violations may lead to fabrication failures, including yield degradation, electrical shorts or opens, electromigration-induced issues, and parasitic mismatches. Moreover, DRC compliance is essential for compatibility with downstream fabrication steps such as photomask generation, optical lithography, and chemical-mechanical planarization (CMP), safeguarding the yield and fidelity of the final IC.

**Circuit-Level Layout Guidelines.** We enforced several topology-aware layout constraints during full-circuit integration to preserve signal integrity and robustness:

- **Inductor-to-inductor spacing:** $\geq 35.0\,\mu$m to mitigate mutual inductive coupling and magnetic interference.

- **Guardring placement:** Sensitive analog blocks are enclosed by N-well or deep N-well guardrings with spacing $\geq 5.0\,\mu$m to suppress substrate noise coupling.

- **Differential pair symmetry:** Differential signal paths are layout-matched to ensure $\Delta L < 0.5\,\mu$m, minimizing mismatch and preserving phase balance.

**DRC Constraints and Layer Definitions.** Table 14 summarizes the DRC constraints applied to key analog components across relevant process layers. Table 15 provides the abbreviations used for metal, contact, and via layers in the 45 nm CMOS process.

Table 14: Design rule constraints for key analog components in 45 nm CMOS.

| Component | Layer | Physical Constraint | Symbol | Value | Unit |
|---|---|---|---|---|---|
| MIM Capacitor (QT, LD, VV, OB) | QT/LD | Minimum Cap Width | $W_{\text{MIN}}$ | 6.05 | $\mu$m |
| | QT/LD | Maximum Cap Width | $W_{\text{MAX}}$ | 150.0 | $\mu$m |
| | QT/LD | Cap Length | $L$ | 6.0 | $\mu$m |
| | VV | VV Square Size | VV_SIZE | 4.0 | $\mu$m |
| | VV | VV Spacing | VV_SPACE | 2.0 | $\mu$m |
| | VV | VV to Edge Spacing | VV_EDGE_MIN | 1.0 | $\mu$m |
| Resistor (RX, CA, M1) | RX | Minimum Width | $W_{\text{MIN}}$ | 0.462 | $\mu$m |
| | RX | Maximum Width | $W_{\text{MAX}}$ | 5.0 | $\mu$m |
| | RX | Minimum Length | $L_{\text{MIN}}$ | 0.4 | $\mu$m |
| | RX | Maximum Length | $L_{\text{MAX}}$ | 5.0 | $\mu$m |
| | CA | Contact Size | CA_SIZE | 0.06 | $\mu$m |
| | CA | Contact Spacing | CA_SPACE | 0.10 | $\mu$m |
| | CA | CA to Edge Spacing | CA_EDGE | 0.11 | $\mu$m |
| Inductor (M3) | M3 | Minimum Width | M3_W_MIN | 2.0 | $\mu$m |
| | M3 | Maximum Width | M3_W_MAX | 20.0 | $\mu$m |
| | M3 | Minimum Spacing | M3_S_MIN | 2.0 | $\mu$m |
| Grid | All Layers | Minimum Grid | Min_Grid | 0.005 | $\mu$m |

Table 15: Process layer abbreviations in the 45 nm CMOS design flow.

| Layer Name | Description |
|---|---|
| **RX** | Resistor implant or diffusion layer used to define integrated resistor geometries. |
| **CA** | Contact layer forming vias between diffusion/poly and the first metal layer (M1). |
| **M1** | First metal layer, typically used for local interconnects and resistor terminals. |
| **M3** | Third metal layer, used for wider routing tracks and planar inductor layouts. |
| **QT** | Top metal plate in MIM capacitor structures, providing the upper electrode. |
| **LD** | Lower metal plate in MIM capacitor structures, acting as the bottom electrode. |
| **VV** | Via layer connecting different metal layers, especially in capacitor and dense routing regions. |
| **OB** | Opening/blocking layer used to define restricted zones, often to exclude metal or for CMP mask clarity. |

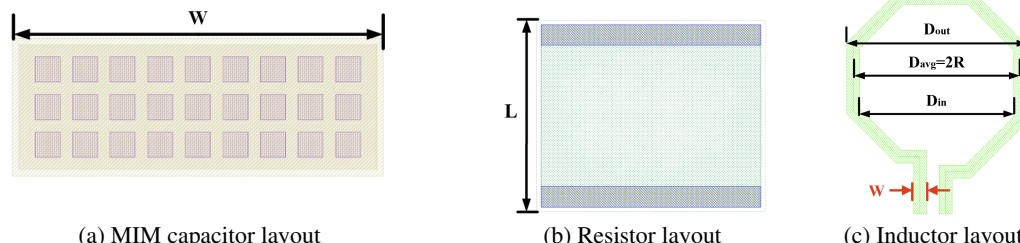

| (a) MIM capacitor layout | (b) Resistor layout | (c) Inductor layout |

Figure 14: Layout views of passive components. (a) MIM capacitor with metal-insulator-metal stack. (b) Resistor layout with matching geometry. (c) Spiral inductor with octagonal turns for optimized area and Q-factor.

## E.2 MIM Capacitor Capacitance Model

The total capacitance $C_N$ of a metal-insulator-metal (MIM) capacitor is modeled as:

$$C_N = C_a \cdot L \cdot W + C_p \cdot 2 \cdot (L + W) \quad \text{[fF]}$$

where $L$ and $W$ are the layout length and width in $\mu$m, $C_a$ is the area capacitance density, and $C_p$ is the fringing field contribution per unit length. This model includes both area and perimeter contributions to more accurately reflect layout-dependent capacitance in IC design (see Figure 14(a)).

**1. Area Capacitance Term:**    $C_a \cdot L \cdot W$

*Physical Concept:* This term represents the primary (parallel-plate) capacitance formed between the overlapping top and bottom metal layers. It arises from the uniform electric field across the dielectric.

*Layer Physics Explanation:*

- $L \cdot W$ corresponds to the overlap area of the plates.
- $C_a = 0.335\,\text{fF}/\mu\text{m}^2$ is the area capacitance density, derived from:
    - Dielectric permittivity $\varepsilon$ of the insulating material.
    - Dielectric thickness $d$, with $C \propto \varepsilon/d$.

**2. Perimeter (Fringing) Capacitance Term:**    $C_p \cdot 2 \cdot (L + W)$

*Physical Concept:* This term models fringing fields at the plate edges, contributing additional capacitance—particularly relevant in small geometries.

*Layer Physics Explanation:*

- $2 \cdot (L + W)$ is the physical perimeter of the capacitor.
- $C_p = 0.11\,\text{fF}/\mu\text{m}$ accounts for the fringing field contribution per unit length.

**Summary:** This composite model enables accurate estimation of MIM capacitance by capturing both parallel-plate and fringing effects. The constants $C_a$ and $C_p$ are typically calibrated using process-specific measurements or electromagnetic simulations.

For a fixed capacitor length $L = 20\,\mu$m and width $W \in [6.05, 150.0]\,\mu$m, the layout-aware capacitance is approximated by:

$$\boxed{C \approx 6.92\,W + 4.4 \quad \text{[fF]}} \tag{4}$$

The corresponding bounding area is estimated from the component's geometric envelope:

$$\boxed{\text{Bounding\_Area} = 22W + 44 \quad [\mu\text{m}^2]} \tag{5}$$

## E.3 N$^+$ Silicided Polysilicon Resistor Model

The resistance of a layout-defined resistor implemented using the `ndslires` layer is modeled as:

$$R = R_s \cdot \frac{L}{W + \Delta W} + 2R_{\text{end}} + \delta \quad [\Omega]$$

*Physical Concept:* This structure uses heavily doped N⁺ polysilicon overlaid with a silicide layer to reduce resistance. Current flows laterally through the poly-silicide film (see Figure 14(b)), and resistance is shaped by the aspect ratio of the layout as well as process-dependent corrections.

*Layer Physics Explanation:*

- $R_s = 17.6\,\Omega/\square$ (ohm per square) is the sheet resistance of the silicided poly layer.
- $W = 5.0\,\mu\text{m}$ is the drawn width; $\Delta W = 0.048\,\mu\text{m}$ accounts for process-induced width bias.
- $L$ is the drawn resistor length.
- $R_{\text{end}} = 1\,\Omega$ models terminal resistance due to contact diffusion and current crowding.
- $\delta = 0.917\,\Omega$ accounts for residual layout-dependent parasitics.

**Summary:** The empirical layout relation used in parameterized generation is:

$$\boxed{R \approx 3.5007 \cdot L + 2.917 \quad [\Omega]} \tag{6}$$

This model is valid for $L \in [0.4,\ 5.0]\,\mu\text{m}$ with fixed width $W = 5.0\,\mu\text{m}$. The estimated layout area based on bounding box dimensions is:

$$\boxed{\text{Bounding\_Area} = 5.2L + 8.362 \quad [\mu\text{m}^2]} \tag{7}$$

### E.4 Octagon Spiral Inductor Model

*Physical Concept:* Accurate modeling and layout optimization of planar spiral inductors are critical in analog circuit design. Inductor performance is highly sensitive to parasitic elements, achievable quality factor ($Q$), and layout constraints imposed by process design rules. To support accurate performance prediction and inform layout choices, we adopt a modified power-law model that expresses inductance as a function of key geometric parameters. The model is validated against empirical measurements and shows strong agreement with classical analytical formulations.

Numerous classical formulations relate inductance to geometric factors such as the number of turns, average diameter, trace width, and inter-turn spacing. Among these, the compact closed-form expressions in *RF Microelectronics textbook* [30] are widely adopted for their balance of simplicity and accuracy. Building on this foundation, we adopt a reparameterized monomial model that better fits our empirical measurement data:

$$L = 2.454 \times 10^{-4} \cdot D_{\text{out}}^{-1.21} \cdot W^{-0.163} \cdot D_{\text{avg}}^{2.836} \cdot S^{-0.049} \quad [\text{nH}]$$

*Layer Physics Explanation:*

- $D_{\text{out}} = 2(R + \frac{W}{2})$ is the outer diameter,
- $D_{\text{in}} = 2(R - \frac{W}{2})$ is the inner diameter,
- $D_{\text{avg}} = (D_{\text{out}} + D_{\text{in}})/2 = 2R$ is the average diameter,
- $R$ is the radius in μm,
- $W$ is the trace width in μm,
- $S$ is the spacing in μm.

Table 16: Measured inductance for one-turn inductors with fixed $W = 10\,\mu\text{m}$ and $S = 0.0\,\mu\text{m}$

| $R$ (μm) | 30 | 40 | 50 | 60 |
|---|---|---|---|---|
| $L$ (nH) | 0.123 | 0.170 | 0.220 | 0.276 |

This expression is calibrated using measured data from a series of one-turn inductors fabricated with varying radius ($R$), while keeping the trace width fixed at $W = 10\,\mu\text{m}$ and spacing at $S = 0.0\,\mu\text{m}$. Table 16 summarizes the measured inductance values used for model fitting.

**Summary:** With $W$ and $S$ fixed, inductance simplifies to:

$$\boxed{L \approx 2.337 \times 10^{-3} \cdot R^{1.164} \quad [\text{nH}]} \tag{8}$$

The bounding area is estimated by:

$$\boxed{\text{Bounding\_Area} = 4R^2 + 108R + 440 \quad [\mu\text{m}^2]} \tag{9}$$

The performance of on-chip inductors is fundamentally influenced by layout-dependent factors such as trace width, metal thickness, and inter-turn spacing. Increasing the trace width ($W_{\text{ind}}$) reduces series resistance by enlarging the conductor's cross-sectional area, thereby improving the quality factor, $Q = \omega L / R_{\text{series}}$. However, wider traces also increase parasitic capacitance to adjacent turns and the substrate, which lowers the self-resonance frequency.

Metal thickness ($H_{\text{ind}}$) also plays a crucial role in minimizing ohmic losses. At high frequencies, current is confined near the conductor surface due to the skin effect. For copper at $25\,\text{GHz}$, the skin depth $\delta$ is approximately $0.41\,\mu\text{m}$; thus, using a metal layer thicker than $4\delta$ (i.e., $1.6\,\mu\text{m}$) ensures efficient current flow. However, increasing thickness beyond this threshold yields diminishing returns in $Q$ due to saturation in current penetration.

Turn-to-turn spacing ($S$) affects both inductance and quality factor ($Q$). Tighter spacing enhances magnetic coupling, thereby increasing inductance density. However, it also intensifies capacitive coupling and dielectric losses—particularly in modern CMOS processes with high-$k$ inter-metal dielectrics—which can degrade $Q$. Conversely, excessive spacing reduces inductance without providing a proportionate benefit in loss reduction. As a result, one-turn spiral inductors are commonly favored in RF design due to their low series resistance, minimized parasitics, and improved modeling predictability.

These insights guided our design choices for layout-aware inductor implementation. To balance the competing demands of $Q$ optimization, parasitic control, and DRC compliance, we implemented inductors using Metal 3 and set $W = 10\,\mu\text{m}$ as the default trace width. This width offers a low-resistance path that enhances $Q$ while maintaining manageable parasitic capacitance and sufficient pitch for lithographic reliability. Metal 3 was selected for its favorable trade-off between thickness and routing density—it is thick enough to mitigate skin-effect losses at high frequencies while offering sufficient flexibility for compact layout integration.

The implemented spiral inductor geometry is shown in Figure 14(c). Table 17 summarizes the DRC-compliant tuning ranges, estimated layout areas, and decomposition strategies for single-cell passive components in our layout library.

Table 17: Single-cell passive component limits based on DRC and associated layout area costs.

| Component | Tunable Variable | Value Range | Area Range | Decomposition Rule |
|-----------|------------------|-------------|------------|--------------------|
| Resistor | Length $L$ | $4.32$–$20.42\,\Omega$ | $10.44$–$34.36\,\mu\text{m}^2$ | Series if > max, parallel if < min |
| Capacitor | Width $W$ | $46.32$–$1042.4\,\text{fF}$ | $176$–$3344\,\mu\text{m}^2$ | Parallel if > max, series if < min |
| Inductor | Radius $R$ | $\geq 0.1\,\text{nH}$ | $\geq 5640\,\mu\text{m}^2$ | Continuous radius scaling |

## E.5 Layout Examples of Synthesized Circuits

To illustrate the correspondence between schematic and layout representations, we present three synthesized circuits: DBAMixer, IFVCO, and DLNA, shown in Figures 15, 16, and 17, respectively.

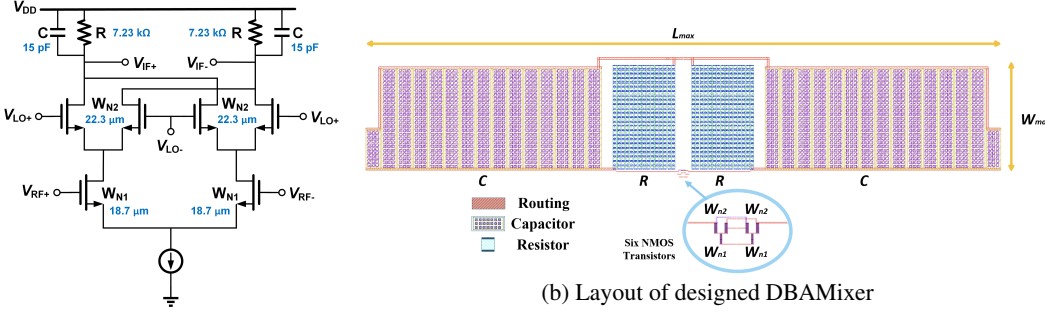

(a) Designed DBAMixer schematic

(b) Layout of designed DBAMixer

Figure 15: Stage 3 results for a synthesized DBAMixer. The schematic (a) reflects optimized parameters to meet the target specification. The layout (b) is DRC-compliant and physically realizable. The final design achieves a mean relative error of 0.2% compared to the target performance.

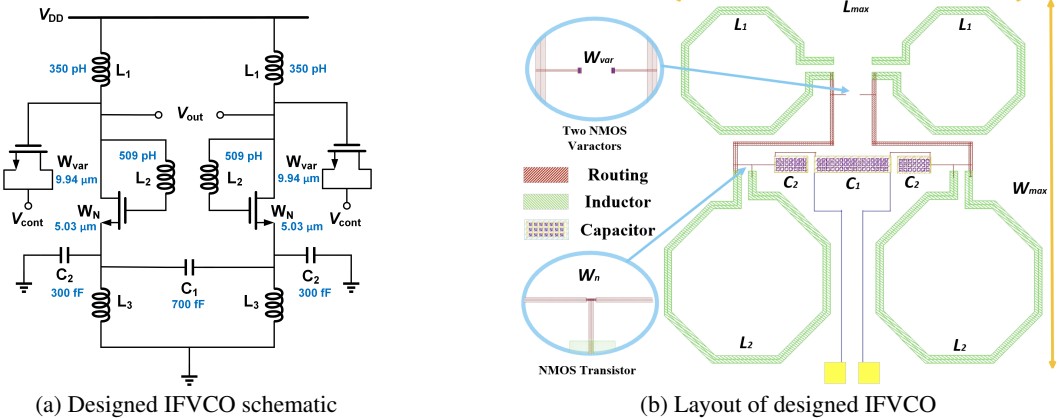

(a) Designed IFVCO schematic

(b) Layout of designed IFVCO

Figure 16: Stage 3 results for a synthesized IFVCO. The schematic (a) reflects optimized parameters to meet the target specification. The layout (b) is DRC-compliant and physically realizable. The final design achieves a mean relative error of 1.3% compared to the target performance.

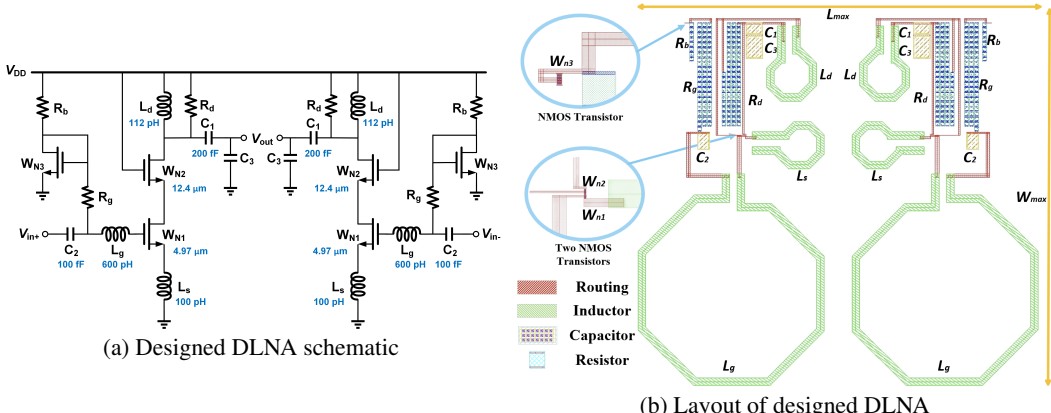

(a) Designed DLNA schematic

(b) Layout of designed DLNA

Figure 17: Stage 3 results for a synthesized DLNA. The schematic (a) reflects optimized parameters to meet the target specification. The layout (b) is DRC-compliant and physically realizable. The final design achieves a mean relative error of 5.0% compared to the target performance.

In the IFVCO example, the inductor labeled $L_3$ functions as an RF choke and is excluded from the on-chip layout due to its large area requirement. Instead, it is intended for off-chip implementation at the PCB level and connected to the die via wire bonding. This external connection is indicated by the yellow pad in Figure 16(b), which serves as the wire-bonding interface.

Since the current stage of system lacks automated routing, all interconnects in the layout were manually drawn to ensure accurate correspondence with the schematic connectivity. These examples demonstrate that synthesized circuit parameters can be successfully translated into DRC-compliant, physically realizable layouts, bridging the gap between high-level optimization and tapeout-ready design.

## F Practical Considerations and Limitations

### F.1 Training and Inference Efficiency

Although our codebase supports GPU acceleration, all experiments in this work—excluding initial dataset generation—were conducted entirely on a MacBook CPU. This highlights the efficiency and accessibility of the FALCON pipeline, which can be executed on modest hardware without specialized infrastructure. Our MLP and GNN models contain 207k and 1.4M trainable parameters, respectively, with memory footprints of just 831 KB and 5.6 MB.

In Stage 1, the MLP classifier trains in approximately 30 minutes with a batch size of 256 and performs inference in the order of milliseconds per batch. Stage 2's GNN model takes around 3 days to train on the full dataset using the same batch size and hardware. Fine-tuning on an unseen topology (e.g., RVCO) using ~30,000 samples completes in under 30 minutes.

In Stage 3, the pretrained GNN is used without retraining to perform layout-constrained parameter inference via gradient-based optimization. Inference is conducted one instance at a time (batch size 1), with typical runtimes under 1 second per circuit. Runtime varies based on the convergence threshold and circuit complexity but remains below 2–3 seconds in the worst case across the full test set.

A solution is considered successful if the predicted performance meets the target within a specified relative error threshold. While tighter thresholds (e.g., 5%) improve accuracy, they require more optimization steps—particularly over large datasets. As a result, both success rate and inference time in Stage 3 are directly influenced by this tolerance, which can be tuned based on design fidelity requirements.

## F.2    Limitations

This work focuses on a representative set of 20 curated analog topologies spanning five circuit families. Consequently, the topology selection stage is limited to suggesting only among the designs present in the training set and cannot synthesize novel circuits. A natural future direction is to either extend the training library to a broader set of topologies or replace the classifier with a generative model capable of directly proposing new netlists conditioned on input specifications. In contrast, the GNN-based forward modeling stage is designed to operate on arbitrary circuit graphs and has already demonstrated strong generalization to unseen architectures (e.g., RVCO), indicating that no modification to this stage is required to support novel circuits.

Beyond topology considerations, the dataset is constructed at a fixed operating frequency of 30 GHz, which ensures consistency across circuit families but constrains frequency generalization. Although the framework can, in principle, extend to other operating points—for example, the voltage amplifier topologies already demonstrate scalability across varying gain–bandwidth trade-offs—systematic validation across diverse frequency bands is beyond the scope of this work. Extending the dataset to cover multiple operating frequencies, or incorporating frequency as an explicit conditioning variable during training, represents an important direction for broadening applicability.

Finally, the differentiable layout model in FALCON captures parasitic effects through analytical approximations of passive components, which is effective for guiding parameter optimization within the learning framework. However, this approach does not fully replace electromagnetic (EM) simulations or post-layout verification, and electromigration constraints are not explicitly incorporated. Incorporating learned parasitic estimators, EM-informed models, and reliability constraints, therefore, remains an important extension toward bridging schematic-level optimization and silicon-proven robustness. In addition, all interconnect routing in the current flow is performed manually to ensure precise control over parasitic management and DRC compliance. While this provides accuracy for the studied designs, it limits scalability for more complex circuits, motivating future integration with automated analog routing tools.

