# OpenReview forum: "FALCON: An ML Framework for Fully Automated Layout-Constrained Analog Circuit Design"
_NeurIPS.cc/2025/Conference — NeurIPS 2025 poster_

### Official Review · Reviewer_Ae6i · 2025-07-01

**Clarity:** 4
**Significance:** 3
**Originality:** 3
**Rating:** 4
**Confidence:** 4

**Summary:**

The paper proposes a unified framework for synthesis of Analog designs. It is aimed at addressing the inverse problem that is given a target specification, design a circuit that meets the spec. The approach involves three steps: a) Topology selection: For a given target performance, choose an optimal topology amongst a fixed set of 20 topologies using a trained MLP classifier b) Performance prediction: For a given topology and parameters, predict the circuit performance using a learned GNN based forward model. c) Parameter inference: Given the target spec and topology, use gradient descent over the learned forward model to find the optimal parameters that meet the requirement.

**Questions:**

( I am willing to re-consider scores if some of the following are addressed, especially comparison with prior art)


1. It would be helpful if the method could handle specification ranges instead of fixed targets, since that’s how design specs are usually defined in practice.
2. Allowing designers to indicate which specs matter more—or where trade-offs are acceptable—could make the tool more usable in real design workflows. One way to achieve this would be to use weighted inputs
3. The paper would benefit immensely from comparison with existing work. While no complete framework exists for direct comparison, each component (topology selection, performance prediction, parameter inference) should be compared against relevant existing methods referenced in the supplementary material (Table 3).
4.  The paper would benefit from ablation studies on layout cost to access its impact, multi-start to evaluate sensitivity local minima, and a study on parameter initialization to understand its effect on design quality.
5.  In addition to mean error statistics, it would also be interesting to see the maximum error stats for all three steps.

*Minor:*
1. Confusion matrix in Fig.3(b) is not complete. Can be added to supplementary material.
2. It would be good to include the discussion on limitations in practical considerations/generalization of this approach in the main paper rather than supplementary.

**Ethical Concerns:**

["NO or VERY MINOR ethics concerns only"]

**Final Justification:**

This work is a step forward towards using ML automated analog synthesis, a difficult and relatively unexplored problem. After reconsidering the authors rebutal and my fellow reviewers comments, I have decided to raise the score to 4.

**Limitations:**

Yes

**Quality:**

3

**Strengths And Weaknesses:**

**Strengths:**  	The paper is well-structured and clearly written. It addresses the challenging problem of automating Analog design from specifications. The approach combines topology selection with parameter inference while maintaining structural fidelity. The inclusion of early layout-aware optimization adds value.

**Weakness:**      The paper lacks comparison with existing ML/non-ML based techniques, making it difficult to assess relative performance, efficiency and generalizability against prior approaches.

The method uses fixed specification targets and a limited set of topologies, which reduces flexibility and scalability. It is also unclear how practical the approach would be for experienced analog designer, given the lack of support for guiding trade-offs or incorporating domain expertise.

---

> ### Author Rebuttal · Authors · 2025-07-31
>
> We thank the reviewer for the constructive feedback. We greatly appreciate the recognition of our contributions and address the concerns below.
>
> **Weaknesses:**
> 1. We thank the reviewer for emphasizing the importance of comparison to prior work. Building on Table 3, we provide additional clarification and results to address the reviewer’s concern in more depth:
> - **Topology Selection:** Although prior work includes topology generation/selection, their problem formulation differs fundamentally from ours. These methods aim to construct valid schematics, often without explicit consideration of specifications. In contrast, FALCON frames topology selection as a performance-driven classification task over a fixed library of mm-wave circuits. Moreover, none of the prior work supports the RF frequency, which significantly impacts circuit behavior and design constraints. Therefore, a direct performance comparison is not feasible.
> - **Performance Prediction:** Among existing approaches, Deep-GEN is the only publicly available method. However, their model excludes key components like inductors and does not support AC performance metrics such as bandwidth, both of which are essential in our mm-wave designs. Additionally, similar to AutoCkt, the released code is incomplete and cannot be used to produce results on our dataset. Thus, a quantitative comparison was unfortunately not feasible.
> - **Parameter Inference:** We identified AICircuit and Krylov et al. as the directly comparable baselines. While both works explore similar modeling strategies, AICircuit offers five distinct models and aligns more closely with our goal in terms of Cadence simulation and circuit scope. To provide a fair comparison, we evaluated AICircuit on our held-out topology (RVCO) and compared it to the FALCON fine-tuned model on RVCO. There are several key distinctions between FALCON and AICircuit:
>    - FALCON integrates a layout loss during parameter inference, enabling practical design refinement. In contrast, AICircuit does not incorporate layout modeling during inference.
>    - FALCON uses a single model trained across 19 topologies, enabling generalization and zero-shot inference on new topologies. AICircuit, by design, requires training a separate model for each topology, since its predictors are tied to topology-specific feature spaces.
>    - FALCON's inference quality is governed by a user-defined error threshold, which makes accuracy tunable. In contrast, AICircuit performs direct regression with no adjustable convergence mechanism.
>
> The relative error results on RVCO are summarized below:
> |Method | Rel Err (%) |
> |-|:-:|
> AICircuit-kNN|4.46|
> AICircuit-SVR|4.09|
> AICircuit-RF|3.91|
> AICircuit-Transformer |3.56|
> AICircuit-MLP|2.42|
> FALCON|2.87|
>
> While AICircuit-MLP achieves slightly lower error under these settings, we note that FALCON enables generalization to unseen topologies, layout optimization, and tunable inference precision. Indeed, lowering our termination threshold would yield improved relative error below MLP, but we maintained consistency with the main paper settings for transparency. We will include this comparative study in the revised version of the paper.
>
> We hope these clarifications and additional evaluations address the reviewer’s concerns. If the reviewer has specific methods they would like us to include, we are happy to consider those as well.
>
> 2. We thank the reviewer for highlighting this important point. We address the limitations regarding fixed targets and trade-offs in our responses to Q1 and Q2. Regarding the limited topology set, FALCON currently operates on a library of expert-designed topologies to ensure physical validity, layout feasibility, and manufacturability. We agree that integrating generative topology methods represents a promising direction for future work.
>
> **Questions:**
> 1. We thank the reviewer for this valuable suggestion. We agree that extending FALCON to support interval-based reasoning is an important direction. Our framework can already be adapted to handle ranges in the key stages that require them:
>  - **Topology Selection:** Although our MLP classifier is trained on fixed targets, it can still support interval-based queries. A simple approach is to input the midpoint of each performance interval or to sweep across the valid range and select the topology that is most frequently predicted across different samples. This preserves the benefits of supervised training while adapting to flexible specs.
> - **Performance Prediction:** This stage maps a given topology and parameter vector to a predicted performance vector. It does not need to process targets directly. Instead, it acts as a surrogate model used in Stage 3 and is independent of how the final loss is defined. Therefore, no changes are needed to this component.
> - **Parameter Inference:** In this stage, the GNN is used as a predictor model to redefine the optimization loss to accommodate performance intervals. We have implemented an alternative loss that supports input spec intervals, where each masked performance may have lower and/or upper bounds (finite or infinite). The loss penalizes deviations from these bounds as follows:
>
> $$L_{\text{interval}} = \sum_{i \in \text{masked}}
> \begin{cases}
> \max(0, \hat{y}_i - y^{\text{upper}}_i) & \text{if upper bound} \\\\
> \max(0, y^{\text{lower}}_i - \hat{y}_i) & \text{if lower bound}
> \end{cases}$$
>
> This masked interval loss replaces the masked MSE used and will be included in our final code release. While our current evaluation adopts fixed targets for consistency, we acknowledge that real-world use cases are often goal-dependent. We welcome the reviewer’s feedback on how best to evaluate interval-constrained optimization and would be happy to include additional results in the paper if desired.
> 2. We appreciate the reviewer’s suggestion. We agree that enabling designers to express trade-offs is valuable in practical design workflows.
> - **Topology Selection:** Since this stage uses an MLP mapping fixed performance vectors to topology classes, incorporating weights directly into its input space is nontrivial. Exploring approaches to enable trade-off-aware classification remains an open research direction, and we welcome further ideas from the community.
> - **Performance Prediction:** This model does not directly interact with specification weights.
> - **Parameter Inference:** We have implemented a weighted loss function for this stage. Users can now provide per-specification weights, and the loss function scales each performance error accordingly during optimization. This allows the reasoning process to prioritize certain metrics over others, supporting designer-driven trade-offs. The type of loss, masked MSE, interval, or weighted, can now be selected via an input argument to the script. We will release this implementation in the supplementary material. While evaluating performance under arbitrary user-defined weights is inherently application-dependent, we welcome any suggestions the reviewer may have regarding evaluation protocols or result presentation.
> 3. We thank the reviewer for raising this important point. As discussed in the weakness section, while direct comparisons are not always possible due to differences in problem formulation, tool availability, or supported circuit types, we have provided clarification on relevant related work and included new results where feasible (comparing our stage 3 to AICircuit).
> 4. We thank the reviewer for these excellent suggestions. We have implemented several experiments to address these points:
> - **Layout Cost:** To isolate the contribution of layout-aware optimization, we perform an ablation study where the layout loss weight λ is set to zero. We report the resulting success rate and mean rel error. Since our optimization loop includes hard constraints on layout feasibility (e.g., area exceeding a threshold causes the candidate to be rejected), reporting fine-grained area differences is less informative: minor improvements do not affect success or feasibility. Nevertheless, we remain open to suggestions on additional layout-related metrics the reviewer would like to see.
> - **Initialization:** We will include a study on the effect of initialization noise variance and type. Since our inference process uses a fixed optimization threshold to determine success, increasing variance affects convergence speed, rather than overall success rate or rel error. Given that the inference completes in under 1sec/sample, even with additional steps, this added cost is negligible in practice. We will report detailed results for different noise types and variances in the appendix.
> We hope these additions address the reviewer’s concerns and are happy to incorporate further evaluations based on reviewer feedback.
> 5. We thank the reviewer for this helpful suggestion. We will include maximum error statistics for the regression stages in the main paper. For Stage 2, the maximum rel error was 42.4%, and for Stage 3, the maximum rel error was 32.1%. The latter value is influenced by the optimization termination threshold used in our experiments: lower thresholds yield lower maximum errors. Note that Figure 5 shows the 95th percentile errors, as stated in the caption. For topology selection, maximum error is not meaningful; however, the lowest per-class accuracy was 93.9%, corresponding to a maximum misclassification rate of 6.1%.
>
> Minor:
> 1. The omitted classes in Fig. 3(b) correspond to those with 100% classification accuracy: no samples from these classes were misclassified, and no other classes were incorrectly classified as them. We will include the full confusion matrix in the appendix.
> 2. We acknowledge that placing the Limitations section in the supplementary material may have made it less visible. If the paper gets accepted, we will move this section to the extra page of the main paper and expand it to incorporate the additional limitations raised by the reviewers.

---

> > ### Comment · Reviewer_Ae6i · 2025-08-07
> >
> > Thanks for your detailed rebuttal. After reconsidering the rebuttal and the paper, I have decided to increase the score to 4.

---

> > > ### Author Response · Authors · 2025-08-08
> > >
> > > Thank you very much for reconsidering your score and for your helpful suggestions during the review process. We sincerely appreciate the opportunity to address your concerns and are glad the clarifications proved helpful. We would be glad to address any remaining questions or suggestions you may have.

---

### Official Review · Reviewer_fujS · 2025-07-01

**Clarity:** 4
**Significance:** 3
**Originality:** 3
**Rating:** 5
**Confidence:** 1

**Summary:**

This work introduces a fully automated framework, FALCON, for designing analog circuits with the aid of machine learning. While various ML-based approaches have been proposed for analog circuit design, they typically focus on isolated steps of the design process—such as topology generation, parameter sizing, or schematic-level performance evaluation.
With three core components—a lightweight MLP that selects circuit topology, a GNN that maps topology to performance metrics, and gradient-based optimization of the design parameters—FALCON enables full automation of the entire design process.
Finally, through training and evaluation leveraging a dataset of one million analog mm-wave circuits, FALCON demonstrates strong performance in topology inference, performance prediction, and efficient layout-aware design.

**Questions:**

- **Q1.** This reviewer does think that one unified framework is a meaningful contribution. However, this reviewer is curious about what differences there would be compared to simply concatenating the known approaches for each step. What would be the benefit of a unified framework over using established methods for each procedure individually? Are there aspects that are qualitatively more challenging when developing a unified framework, as opposed to just applying known approaches?

- **Q2.** Table 1 shows that the classification task for topology selection can be performed with a high score, even using a simple MLP. The scores seem notably high, and this reviewer is curious whether there is any risk of overfitting. To the best of this reviewer's understanding, the evaluation is performed on pre-labeled data. However, the ultimate goal of topology selection is to build a proper circuit that fits a specific purpose, not merely to classify pre-labeled data. What is the justification for assuming that a high classification score will lead to good topology selection for new tasks? Or is there a procedure to adjust the topology in a later step? Also, it seems the classification is limited to 20 topologies. This reviewer is curious whether these 20 topologies are general enough to support the design of a wide range of new circuits.


- **Q3.** The focus of the framework suggested in this paper seems to be analog/RF circuits, but this reviewer feels the overall structure is flexible enough to handle circuits for different tasks. For example, there is a line of work on designing electrical circuits to solve convex optimization problems or to design new convex optimization algorithms. However, if we want to consider circuits for different tasks, I believe the concrete components of the framework would need to change. For example, the topology classification based on an analog/RF circuit dataset should be replaced with another component. Still, it seems possible to retain the idea of updating parameters with gradient descent by modifying the loss function. Do you think the structure of your framework can be adapted to design circuits for different tasks, with appropriate replacements of the components? Or do you think the required changes would be almost equivalent to developing an entirely new framework? Since this question may be outside the scope of your research, the reviewer would appreciate any high-level thoughts the authors are willing to share, if any.

**Ethical Concerns:**

["NO or VERY MINOR ethics concerns only"]

**Final Justification:**

Based on the reasons stated in the Strengths section and after reading the additional discussions during the rebuttal period, I believe this paper deserves to be published in NeurIPS.

**Limitations:**

Yes.

**Paper Formatting Concerns:**

No issues.

**Quality:**

4

**Strengths And Weaknesses:**

**Strengths**

- FALCON is an end-to-end, scalable and modular ML framework for designing analog and RF circuits.
Unlike other prior works, FALCON provides full automation on the entire process.

- To the best of the reviewer's understanding, this paper  utilizes a dataset constructed by the authors themselves, which in itself is a commendable contribution.

- The paper is clearly written and well organized, which makes the high-level idea easy to understand. This clarity is a notable strength of the paper.


**Weaknesses**

- This reviewer has a few questions and minor concerns but couldn’t find any significant weaknesses.

---

> ### Author Rebuttal · Authors · 2025-07-31
>
> We sincerely thank the reviewer for their thoughtful and constructive feedback. We appreciate the encouraging assessment of our work and are pleased that the overall structure and contributions of our framework were well received. Below, we address the reviewer’s insightful questions in detail.
>
> **Questions:**
>
> 1. We thank the reviewer for this thoughtful and important question. While each individual component in our pipeline (topology selection, performance prediction, and parameter inference) can be implemented using isolated methods, FALCON’s unified design offers several unique advantages that go well beyond simple concatenation:
>
> - **Performance-Driven Topology Selection:** To the best of our knowledge, FALCON is the first framework to treat topology selection as a supervised, performance-driven task. Prior works on topology generation or selection focus on structural validity or schematic feasibility, without explicitly optimizing for target performance specifications. This makes them incompatible with an end-to-end design process and difficult to meaningfully integrate into a circuit design pipeline like ours.
>
> - **Layout-Aware Parameter Inference:** FALCON integrates layout awareness directly into the parameter inference stage via a differentiable layout-based loss. In contrast, previous works often treat layout optimization as a disjoint post-processing step, which restricts the achievable design space by fixing parameters beforehand. Our unified approach allows layout constraints to actively shape parameter optimization, improving both feasibility and area efficiency.
>
> - **Generalizable Forward Model:** Unlike prior methods that train separate models per topology, our GNN-based forward model is trained across a diverse set of 19 topologies using a shared representation. This enables *zero-shot inference on unseen topologies (e.g., RVCO in our evaluation), a capability not supported by existing models.* It also supports a wide variety of analog circuit elements, including inductors, capacitors, resistors, pmos/nmos, current/voltage sources, baluns, and ports, making it suitable for nearly all analog circuit blocks.
>
> - **Scalability via Large-Scale Dataset:** Our framework is trained on *an unprecedented dataset of 1 million Cadence-simulated circuits*, which significantly enhances its generalization and robustness. This scale of data *has not been seen in prior works* and allows us to cover a broad spectrum of performance behaviors and circuit topologies.
>
> - **Flexible Optimization Interface:** FALCON’s parameter inference stage is designed to support arbitrary loss functions without requiring retraining. As suggested by other reviewers, we were able to easily implement support for performance intervals, weighted losses, and other designer-specified constraints in less than 30 lines of code. This flexibility is crucial for real-world use and difficult to achieve with rigid standalone models.
>
> - **Seamless Integration and Modularity:** All stages in FALCON share a consistent circuit representation and interface, enabling seamless integration without the need for conversion logic or compatibility hacks. Each module directly consumes the output of the previous stage, ensuring smooth data flow and facilitating end-to-end optimization.
>
> - **Qualitative Challenges in Unified Design:** Designing a pipeline that remains *differentiable, interpretable, and modular across all stages*—while scaling to unseen topologies—requires careful engineering of graph encodings, performance masking, normalization, and objective formulation. These challenges are often overlooked in isolated approaches but are critical in building a fully unified system.
>
> We believe these qualitative and technical advantages distinguish FALCON from a mere aggregation of prior techniques and demonstrate the value of a principled, end-to-end design framework.
>
> 2. We thank the reviewer for this thoughtful question. Below, we address each aspect in turn:
>
> **Risk of Overfitting:** ```Table 1 shows that the classification task for topology selection can be performed with a high score, even using a simple MLP. The scores seem notably high, and this reviewer is curious whether there is any risk of overfitting.```
>
> The high classification accuracy is a consequence of the fact that many circuit topologies exhibit distinct performance behaviors. As shown in the t-SNE visualization (Figure 3(a)), the performance specifications form well-separated clusters across topologies, making the classification problem naturally separable. To assess potential overfitting, we performed an ablation study on the number of MLP layers and hidden sizes, which will be included in the appendix. The 5-layer configuration was selected to balance performance and simplicity, and to avoid overfitting. We also emphasize that the reported results are based on a fully held-out test set, separate from training and validation data.
>
> **Justification for High Accuracy and Post-Selection Optimization:** ```To the best of this reviewer's understanding, the evaluation is performed on pre-labeled data. However, the ultimate goal of topology selection is to build a proper circuit that fits a specific purpose, not merely to classify pre-labeled data. What is the justification for assuming that a high classification score will lead to good topology selection for new tasks? Or is there a procedure to adjust the topology in a later step?```
>
> Each sample in our dataset is labeled with its most suitable topology among 20 candidates, selected based on simulation results from Cadence. Thus, the MLP learns a performance-driven mapping, rather than arbitrary classification. While the topology selection step is discrete, our Stage 3 parameter inference module provides continuous fine-tuning of the design by optimizing component values while preserving connectivity. That is, the structure of the chosen topology is retained, and gradient-based parameter refinement ensures the final design aligns with the target performance. We agree that developing a generative model capable of synthesizing new topologies directly from specs is an exciting future direction.
>
> **Scope of the 20 Topologies:** ```Also, it seems the classification is limited to 20 topologies. This reviewer is curious whether these 20 topologies are general enough to support the design of a wide range of new circuits.```
>
> The 20 topologies span five major analog circuit families—VAs, VCOs, mixers, LNAs, and PAs—and include structural variants to capture diverse performance profiles. While no finite set can cover all possible circuits, this collection includes the most canonical analog blocks. We also highlight that our framework is *modular and extensible*: users can add new topologies and retrain the model as needed. Expanding the training dataset will only enhance the framework’s generalizability, and we view this as an important next step for broader adoption.
>
> 3. We thank the reviewer for this insightful question. We agree that the structure of our framework is conceptually broad and adaptable. If the target application shifts from analog/RF design to a fundamentally different domain, such as circuits designed to solve convex optimization problems, some components of FALCON would indeed need to be modified, but *not redesigned from scratch*.
>
> - **Topology Selection (Stage 1):** Our current MLP classifier is trained to map performance specs to one of 20 pre-defined analog/RF topologies. For a different task, such as designing circuits for optimization solvers, this module would likely need to be replaced with a different selector or a generator tailored to the new family of topologies. That said, the concept of learning a mapping from task specifications to suitable structures still applies, and this stage can be retrained accordingly.
>
> - **Performance Prediction (Stage 2):** This stage is inherently general. Given a new dataset of circuit graphs and their corresponding task-specific performance metrics, our GNN model can be retrained to model the new behavior. The architecture supports arbitrary topologies, components, and performance dimensions/types, making it a reusable backbone.
>
> - **Parameter Inference (Stage 3):** This component is fully adaptable to new domains. The optimization loop is guided by a user-defined loss function, and no retraining is required. For new tasks, one can define a loss appropriate to the desired circuit behavior (e.g., a residual from the solution to a convex problem), and the optimization will adjust parameters accordingly.
>
> In summary, we believe that FALCON's overall structure is flexible and can be adapted to other circuit tasks. The level of modification needed is moderate: the core architecture and methodology remain applicable, but retraining or replacing task-specific components is required. We appreciate the reviewer raising this broader applicability perspective.

---

> > ### Comment · Reviewer_fujS · 2025-08-05
> >
> > This reviewer sincerely appreciates the authors' thorough and thoughtful responses. This reviewer believes this is solid work and will maintain the current score.

---

> > > ### Author Response · Authors · 2025-08-05
> > >
> > > Thank you very much for your kind and supportive feedback. We sincerely appreciate your recognition of the quality of the work. Your encouraging words are truly motivating, and we’re grateful for your thoughtful engagement throughout the review process.

---

### Official Review · Reviewer_TBnj · 2025-07-03

**Clarity:** 3
**Significance:** 3
**Originality:** 3
**Rating:** 5
**Confidence:** 3

**Summary:**

This paper presents an ML framework for automating analogue circuit design. The framework consists of: (1) topology selector module, (2) forward GNN mapper that evaluates the performance of a topology-parameter pair, and (3) differentiable layout-constrained optimisation of circuit parameters. The framework is evaluated on a 1 million dataset of millimetre-wave circuits to demonstrate successful end-to-end analogue circuit design.

**Questions:**

- In Section 3.3 line 165, is the term “reasoning” for describing the optimisation of Eq. (1) the best term to use?
- In Section 4, line 184, regarding the 5-layer MLP, is it possible to ablate this?
- For Figure 3(a), can you provide expanded commentary on the t-SNE visualisation. For instance, could you speculate as to what the strands in the pastel purple cluster (north-most) may mean from a circuit topology standpoint?
- In Section 5, paragraph “Evaluation” (right after line 237), can you repeat evaluation 19 times so that the held-out topology (1 out of 19) is swept across all available topologies? This will help us understand if RVCO is particularly easy (or difficult) to generalise for.

**Ethical Concerns:**

["NO or VERY MINOR ethics concerns only"]

**Final Justification:**

I have read the authors rebuttal and fellow reviewers' comments. Limitations notwithstanding, I believe this work is a solid accept as it advances SOTA in automated analogue design and is likely to seed and inspire further work. The authors are encouraged to consider points raised during the review process in their final manuscript.

**Limitations:**

No. This reviewer suggests considering the following additional limitations:
- The topologies are characterized @30GHz, but this is not sufficient. The topology choice can vary massively depending on the frequency and therefore a design at 2.5GHz is likely to require a different topology, respect to a design at 30GHz
- The analysis of the parasitics is approximate. The authors admit that the next step is the addition of EM modelling to improve accuracy of the parasitic. Simulating a 30GHz RF block without an adequate EM model of the interconnects is unreliable, in this reviewer’s opinion a waste of time. I am also unclear on how the accuracy is calculated. Does it compare the results of the model vs. inaccurate non-EM extracted Spectre sims? If so, the correspondence with real silicon results is likely to be rather weak
- For high-power devices (especially PAs), electromigration must be considered. Especially in advanced nodes, this often drives the transistors and interconnect sizing. Ignoring this can lead to underestimation of the parasitics.

**Paper Formatting Concerns:**

No.

**Quality:**

3

**Strengths And Weaknesses:**

### Strengths:
- Advancing the state-of-the-art of automated analogue circuit design is important and timely
- Releasing a dataset to support further ML-based analogue design research is useful
- Appendices A to E do a good job at supplementing the core treatment for better understanding in this specialised field of research
### Weaknesses:
- From an analogue/RF design standpoint, the work lacks in detail, especially when it presents the results achieved by the Circuit Designer
- In the Evaluation section, the authors present the results of 9500 test instances (nearly 80% success rate and 17.7% relative error); however, these metrics are highly dependent on the circuit specifications fed into the algorithm. This is a key limitation, as the paper doesn’t explore how performance varies with more challenging or non-standard specs.
- Appendix F.2 touches on the limitations of the work, but further discussion of limitations needs to be added. See below.

---

> ### Author Rebuttal · Authors · 2025-07-31
>
> We sincerely thank the reviewer for their thoughtful evaluation and constructive feedback. We appreciate the recognition of FALCON's contributions and the valuable suggestions that will help strengthen the clarity and impact of our work. Below, we address each of the reviewer’s comments in detail.
>
> **Weaknesses:**
> 1. We sincerely thank the reviewer for raising this important point. Given the machine learning focus of the conference, we prioritized ML-related content in the main paper and limited the inclusion of detailed circuit/layout analysis. However, we agree that presenting results from an analog/RF design standpoint is valuable for broader impact. In the final version, we will enrich the discussion with additional circuit-specific interpretations, such as design insights and representative success cases, and would be grateful to incorporate further suggestions from the reviewer.
> 2. We thank the reviewer for highlighting this important limitation. Based on this feedback, we have incorporated a fixed bounding mechanism for each circuit type to prevent users from providing illogical or physically infeasible performance targets. While this issue does not arise in our current evaluation, since all performance targets are derived from expert-designed circuits in our dataset, we believe this additional safeguard will enhance the robustness of FALCON in broader applications and for non-expert users.
>
> To clarify how our system handles target specifications: when a feasible and meaningful performance vector is provided, FALCON uses its learned prior from the 1M-circuit dataset to infer the best corresponding parameters. Naturally, the broader and more diverse the training data, the more generalizable the model becomes. One of the strengths of our framework is its flexibility; FALCON can be retrained or fine-tuned on any analog circuit dataset, enabling the community to extend its capabilities to new specification ranges.
>
> 3. We thank the reviewer for pointing this out. We will incorporate the additional limitations the reviewer raised into our existing Limitations section and, if the paper is accepted, will move this section to the extra page of the main paper to ensure better visibility.
>
> **Questions:**
> 1. We appreciate the reviewer’s comment regarding terminology. We chose the term “reasoning” to emphasize that the optimization process leverages the differentiable forward model to iteratively refine parameters in a manner akin to how a designer reasons about circuit behavior. However, we understand that this may be unconventional in the context of optimization, and we are open to revising the phrasing to more standard terms such as “gradient-based optimization” if preferred.
> 2. We thank the reviewer for this important suggestion. Given the number of topologies and available data points, we conducted an ablation study prior to submission on both the number of layers and the hidden size of the MLP. Although the results were largely similar, with all configurations achieving over 99% total accuracy on the test set, we selected the best-performing and most lightweight configuration to avoid overfitting. We will include the details in the appendix.
> 3. We appreciate the reviewer’s thoughtful observation on t-SNE visualization. The pastel purple cluster (corresponding to topology 9) indeed exhibits a filament-like or strand structure in the t-SNE embedding. Our interpretation is that this topology supports a broader range of performance behaviors, likely due to its architectural complexity or greater configurational diversity in the dataset. This diversity leads to more dispersed feature embeddings, resulting in distinct strand-like subgroups rather than a compact cluster.
>
> This structure may also suggest that the representation space for topology 9 is more sensitive to parameter variations, which causes the learned embeddings to separate more along performance-driven dimensions. We will include this analysis in the final version of the paper to better illustrate how topology complexity manifests in the feature space.
>
> 4. We thank the reviewer for this thoughtful suggestion. To clarify, the evaluation results presented in the main paper are based on a test split that includes all 19 topologies. The held-out evaluation, in which the RVCO topology is excluded from training and only used during fine-tuning and testing, is provided in detail in the supplementary material.
>
> Regarding our choice of RVCO as the held-out topology, we aimed to select a complex and representative topology for a fair evaluation. Among the circuit families, PAs and VCOs are particularly challenging due to their structural complexity and intricate performance relationships [1]. Within this group, we excluded topologies that were either too simplistic (e.g., ColVCO, CCVCO) or too structurally unique (e.g., DPA, which is the only topology involving transformer structures modeled as mutually coupled inductors). RVCO offered a balanced choice, complex but not uniquely structured, making it a strong candidate for assessing generalization. In addition, RVCOs are widely used for power generation at mm-wave and THz frequencies [2], further supporting their relevance as a held-out topology.
>
> Repeating this held-out evaluation for all 19 topologies would require training 19 new foundation models, which is computationally expensive (approximately one week per model). However, we agree that understanding generalization across different complexity levels is valuable, and we will add a few additional held-out experiments on diverse topology types to provide further insight.
>
> **Limitations:**
>  - We thank the reviewer for this thoughtful comment. We fully agree that the choice of circuit topology is frequency-dependent and that a topology suitable at 30 GHz may not be appropriate at lower frequencies such as 2.5 GHz. As the reviewer insightfully points out, RF circuits designed for different frequency bands often require distinct sizing strategies, layout considerations, and even architectural choices. However, we would like to mention that lumped components, such as inductors and MOM/MIM capacitors, can still be used at frequencies above 30 GHz if enough caution is taken regarding their self-resonance, quality factor, and parasitics [3].
>
> Moreover, while our current version focuses on 30 GHz circuits, we believe the general pipeline can be extended to broader frequency ranges. In fact, the overall schematic structure of many analog/RF blocks (e.g., voltage amplifiers) remains invariant over moderate frequency shifts, with only the component values or device sizing needing adjustment. To this end, the frequency can be integrated as an input (or target) into the pipeline, enabling future research to handle frequency-aware design, provided sufficient training data is available. We hope that future community efforts can support expanding this dataset across wider operating conditions. We appreciate the reviewer for highlighting this important aspect, and we will explicitly include the fixed-frequency setting as a limitation and discuss how FALCON can be extended to broader frequency ranges in future research.
>
> - We thank the reviewer for this important point. Indeed, the current layout stage uses an analytical parasitic model, which is an approximation. We fully agree that incorporating detailed EM modeling is a necessary next step, particularly for high-frequency designs where interconnect parasitics can have a significant impact. Enhancing layout realism with EM-aware modeling is an active direction we are pursuing.
>
> At present, the error reported in our results reflects validation at the schematic level using Cadence Spectre simulations. In these simulations, transistor models are already post-layout PDK models, while passive components (such as inductors and capacitors) are based on empirical models provided by the foundry PDK. While this approach does not fully capture parasitics introduced by custom layout or interconnect geometry, our past experience comparing simulation to silicon measurements suggests that this setup remains reasonably accurate up to 30 GHz. Nonetheless, we recognize the reviewer’s concern that schematic-level validation without EM extraction may limit real-world fidelity at higher frequencies, and we appreciate the opportunity to highlight this as an important limitation and motivation for future work.
>
> - We thank the reviewer for raising this important point. We fully agree that electromigration is a critical reliability concern, particularly for high-power devices such as power amplifiers. In our current implementation, we adhered to a maximum current density of 24 mA/µm width for thick RF metals, following the guidelines in the 45 nm CMOS PDK. This is comfortably below the PDK-specified limit of 33 mA/µm width. These constraints were enforced by the trace width limits incorporated into our layout generator. In addition, minimum-width DRC rules in our flow help prevent the generation of overly thin interconnects, which further mitigates electromigration risk. That said, we acknowledge that these steps represent only a first-order safeguard. A more comprehensive treatment of electromigration is an important direction for future work, especially for very high-power blocks such as PAs.
>
> **References**
>
> [1] Behzad Razavi. *RF microelectronics*, volume 2. Prentice hall New York, 2012.
>
> [2] O. Momeni and E. Afshari, *High Power Terahertz and Millimeter-Wave Oscillator Design: A Systematic Approach*, in IEEE Journal of Solid-State Circuits, vol. 46, no. 3, pp. 583-597, March 2011, doi: 10.1109/JSSC.2011.2104553.
>
> [3] N. Rostomyan, M. Ozen and P. Asbeck, *Comparison of pMOS and nMOS 28 GHz high efficiency linear power amplifiers in 45 nm CMOS SOI*, 2018 IEEE Topical Conference on RF/Microwave Power Amplifiers for Radio and Wireless Applications (PAWR), Anaheim, CA, USA, 2018, pp. 26-28, doi: 10.1109/PAWR.2018.8310058.

---

> > ### Comment · Reviewer_TBnj · 2025-08-03
> >
> > Dear authors,
> >
> > Thank you for your comments. I have increased my rating to accept.
> >
> > As mentioned above, I believe this work is a solid accept. I encourage you to consider points raised during the review process in your final write-up.

---

> > > ### Author Response · Authors · 2025-08-05
> > >
> > > Thank you very much for your thoughtful and supportive response. We're truly grateful for your time and for increasing your rating. We appreciate your positive assessment of the work and will carefully incorporate the constructive points raised during the review process in the final version.

---

### Official Review · Reviewer_RMej · 2025-07-04

**Clarity:** 3
**Significance:** 3
**Originality:** 2
**Rating:** 5
**Confidence:** 3

**Summary:**

This paper introduces a new, single, and scalable machine learning framework for the entire end-to-end analog circuit design flow, including such tasks as topology selection, parameter inference, and layout-aware optimization. Analog and RF design flows conventionally tend to be manual, iterative, and strongly reliant on designer heuristics. FALCON addresses these limitations by formulating circuit synthesis as an inverse problem: given a target performance specification, the framework learns both the circuit topology and parameter configuration to meet the target.
The system operates in three stages. In the first stage, it demonstrates the performance-directed topology selection by using a light multilayer perceptron. Secondly, it illustrates the performance prediction by a graph neural network that has been trained on Cadence-simulated circuit graphs. Eventually, this stage emphasizes the gradient-based optimization layout-driven with differentiable constraints to parasitic and physical layout rules. Overall, one of the key contributions is to bring layout goals directly into the optimization loop so that schematic and layout concerns can be represented jointly.

**Questions:**

Question 1: Can you incorporate or evaluate a richer parasitic/EM-aware surrogate (e.g., a simple analytic coupling model or pre-trained EM estimator) in your differentiable layout loss? Your current area-only proxy may underestimate critical mm-wave effects, risking larger errors in post-layout verification. The paper can include a small ablation comparing pure area loss vs. an EM-informed loss on a subset of topologies.
Question 2: Have you measured variability across multiple training runs? Reporting only single-run metrics makes it hard to assess stability, especially for large GNNs. Your team should provide mean±std for key metrics (topology accuracy, forward RMSE, inversion error) over several runs.
Question 3: Have you tested applying FALCON to lower-frequency analog or mixed-signal blocks? Broader applicability would greatly enhance the framework’s impact in EDA. At minimum, discuss the modifications needed or outline a small pilot on one low-frequency topology.
Question 4: Given the high training cost, can you provide a “light” version (e.g., subset of 100k samples) or share pretrained weights to lower the barrier for replication? Accessibility to smaller labs/companies is critical for adoption and follow-up research. Your team can release a trimmed dataset + pretrained model checkpoint for at least one topology family.

**Ethical Concerns:**

["NO or VERY MINOR ethics concerns only"]

**Final Justification:**

I keep my "accept" score.

**Limitations:**

No. Suggestions for Improvement: The paper would benefit from an explicit “Limitations” section that summarizes key boundaries of FALCON’s applicability such as its reliance on a fixed topology library, simplified layout surrogate, and exclusive focus on mm-wave blocks so that readers immediately grasp where the approach may fall short.

**Quality:**

3

**Strengths And Weaknesses:**

Strengths：
1. The authors present and publicly release an exceptionally large, high-fidelity dataset of over one million Cadence Spectre simulations spanning 20 expert-designed mm-wave analog/RF topologies. Such an industrial-scale, process-calibrated dataset is exceedingly rare in the academic literature.
2. FALCON integrates three distinct components—performance-driven topology selection, a graph-neural-network forward model, and a differentiable layout cost—into one seamless, differentiable pipeline. This modular design is among the first to jointly handle inverse circuit design and layout constraints in a unified framework. By embedding parasitic effects and soft design-rule penalties directly into the gradient-based optimization via a differentiable layout loss, the method ensures that the inferred parameters meet both electrical performance targets and physical manufacturability requirements, filling a gap left by traditional “simulate-then-layout” or post-hoc layout workflows.
3. FALCON demonstrates few-shot generalization by maintaining high accuracy on an entirely unseen topology, indicating its potential for rapid adaptation to new circuit families with minimal additional data. Furthermore, all sub-modules rely on lightweight MLPs or edge-centric GNNs, and the final inversion stage runs in milliseconds to seconds without invoking time-consuming circuit simulators. This real-time performance aligns with the stringent requirements of practical automated design toolchains, highlighting the framework’s clear path to industrial deployment.

Weaknesses：
1. FALCON assumes a “fixed but extensible” library of expert-designed circuit topologies, which means it cannot autonomously discover or generate novel structures. In contrast to purely generative topology search methods, its capacity for true topological innovation is bounded by the existing template set. This reliance may limit FALCON’s ability to propose entirely new architectures beyond human-curated designs.
2. The differentiable layout cost in FALCON is based solely on simple two-dimensional area calculations for passive components, omitting more complex parasitic couplings, electromagnetic interactions, and multilayer interconnect effects. In high-precision mm-wave designs, these richer physical phenomena can dominate performance, so a purely area-based surrogate may underestimate post-layout discrepancies, potentially widening the gap between predicted and actual behavior.
3. All quantitative results in the paper derive from a single training run with fixed random seeds and there is no reporting of standard deviations, confidence intervals, or other measures of variability across multiple trials. For complex models trained on large-scale data, stability and sensitivity to initialization can materially affect conclusions. The absence of statistical significance metrics makes it difficult to assess the robustness of the reported performance gains.
4. While FALCON covers five key mm-wave circuit type, its evaluation is confined to the mm-wave frequency band. The framework’s generality and performance in lower-frequency analog circuits, mixed-signal systems, or digital-analog co-design scenarios remain untested. This restriction may hinder broader adoption in electronic design automation, where diverse frequency domains and mixed-signal workflows are commonplace.

---

> ### Author Rebuttal · Authors · 2025-07-31
>
> We sincerely thank the reviewer for their thoughtful and constructive feedback. We appreciate the time and care taken to engage with our work, and we are grateful for the encouraging evaluation and insightful suggestions that have helped us improve the paper.
>
> **Weaknesses:**
> 1. We thank the reviewer for raising this important point. FALCON currently operates on a fixed library of expert-designed topologies to ensure physical validity, layout feasibility, and manufacturability. While this limits autonomous discovery of novel architectures, we believe it provides a robust foundation for real-world deployment. We agree that expanding to generative topology search is a promising future direction. In future work, we plan to explore incorporating schematic-level generators that can propose new topologies, which could then be evaluated and optimized within the existing FALCON pipeline.
>
> 2. We thank the reviewer for highlighting this important aspect. Our pipeline is designed to generate near-realistic, manufacturable IC layouts for mm-wave circuits at 30 GHz. In this work, we focused on a layout objective that minimizes the physical area of passive components, which serves as a differentiable proxy in Stage 3. This area-based objective is the default loss used throughout the main experiments. In addition, we have implemented two other layout variants—minimizing interconnect length and minimizing congestion—to explore trade-offs among compactness, signal integrity, and routability. We will include these additional variants as an ablation study in the appendix.
>
> While the current differentiable loss relies on geometric proxies, our layout generator incorporates several physical constraints, such as self-resonance frequency, quality factor, and DRC compliance, to ensure that passive component layouts remain suitable for baseline mm-wave design. We fully agree that geometric loss alone cannot capture the full range of EM effects. Extending the layout cost to incorporate EM-aware modeling represents a valuable future direction, particularly for capturing coupling and loss at higher frequencies.
>
> 3. We thank the reviewer for raising this important point regarding robustness and variability. We have now performed additional experiments to assess the stability of our results across multiple seeds:
>
> - **Topology Selection:** We conducted 5 independent runs with different random seeds. The results, summarized below, show extremely low variance, reflecting the deterministic nature of the classification task given the strong separation between topologies in the performance space:
>
> |Metric|Mean ± 95% CI|
> |-|:-:|
> |Accuracy| 99.57 ± 0.01|
> |Balanced Accuracy | 99.34 ± 0.02|
> |Macro Precision| 99.27 ± 0.01   |
> |Macro Recall| 99.34 ± 0.02|
> |Macro F1| 99.30 ± 0.01|
> |Micro F1| 99.57 ± 0.01|
>
> - **Performance Prediction:** Due to the longer training time per run, we performed 3 runs with different seeds. The mean relative error across 3 random seeds was **8.80 ± 0.34%** (95% confidence interval). Full statistics (including per-performance results) will be included in the final version of the paper. The results show that performance is stable across seeds.
> - **Parameter Inference:** This stage does not involve training and is not sensitive to seed initialization. The only source of randomness is the initial parameter vector for gradient reasoning. However, this has a negligible effect on the final results due to the fixed termination criterion: random noise may alter the number of steps but not the output quality, and the optimization remains highly efficient (under 1 second per sample on CPU). Considering that another reviewer also inquired about the role of initialization in this stage, we will include an ablation study in the final version, analyzing the effect of noise type and variance on convergence and accuracy.
>
> We appreciate the reviewer’s suggestion and will incorporate these robustness analyses into the final paper to strengthen the experimental evaluation.
>
> 4. We sincerely thank the reviewer for raising this important point. We fully agree that extending FALCON to cover lower-frequency analog circuits, mixed-signal systems, and broader co-design scenarios is essential for increasing its practical utility and adoption. While the current version focuses on mm-wave designs, the core architecture of FALCON is adaptable to a wider range of circuit types. Specifically:
> - Our pipeline can naturally incorporate frequency into the design process in two flexible ways. First, frequency can be treated as a user-defined target and concatenated with the performance vector across all stages; in this case, the GNN forward model and MLP predict the corresponding circuit parameters and topology that meet the specified frequency and performance. Alternatively, frequency can be treated as an output of the pipeline, where the gradient reasoning determines the optimal frequency alongside the circuit parameters (the input of the forward GNN model). Although our baseline evaluation is limited to 30 GHz due to the scope of our dataset, this is not a limitation of the framework itself. The pipeline is structurally capable of supporting frequency-aware design given appropriate training data.
> - We chose to focus on mm-wave circuits at 30 GHz for two main reasons: (i) our group’s expertise in RF, mm-wave, and THz design enabled us to select robust and realistic circuit schematics; and (ii) layout effects become increasingly critical at higher frequencies, making layout-aware design especially relevant for mm-wave applications.
>
> That said, we view this as a natural and exciting direction for future work and believe that FALCON’s modular structure makes such extensions feasible with appropriate data and design targets.
>
> **Questions:**
> 1. We thank the reviewer for this insightful suggestion. To mitigate electromagnetic (EM) losses and parasitic coupling in mm-wave and analog layouts, our framework integrates several spatial strategies during placement and routing. These include mandatory spacing rules for interconnects and passive components to reduce both capacitive and inductive coupling.
>
> Specifically, we enforce a minimum spacing of 5 μm between routed signal traces by defining exclusion zones that prevent nearby routing. For passives such as inductors, which are particularly sensitive to mutual coupling at mm-wave frequencies, we impose an edge-to-edge clearance of at least 15 μm. This threshold was determined based on EM simulations showing negligible coupling beyond this distance. These spacing constraints are automatically handled during layout generation and are reflected in the loss computation used for optimization.
>
> To further enhance EM-awareness, we will introduce a vertical coupling constraint: limiting the overlap area between stacked upper metal layers (e.g., M8 and M9) to reduce unintended capacitive interactions between interconnects on different layers. This rule will be incorporated into the layout engine and corresponding loss function. In the final version of the paper, we will include an ablation comparing our current area-based layout loss against this enhanced EM-informed variant, using a representative subset of topologies.
>
> We fully agree that integrating analytic or learned EM surrogates is a valuable direction for future work. While our current implementation does not incorporate such models directly, we view this as a natural extension of the FALCON framework and appreciate the reviewer for highlighting this important point.
>
> 2. We thank the reviewer for raising this point and kindly refer to our response to Weakness 3, where we address training variability and provide statistics across multiple runs.
>
> 3. We thank the reviewer for raising this important point. As noted in our response to Weakness 4, extending FALCON to lower-frequency analog and mixed-signal circuits is a natural and valuable direction for future work. While this version focuses on mm-wave designs, the core framework is flexible and can be adapted to broader frequency ranges and circuit classes.
>
> Among the circuits currently included, our voltage amplifiers closely resemble their low-frequency counterparts in structure and design principles, and may serve as a preliminary example of applicability in that domain. That said, we agree that a dedicated evaluation on diverse low-frequency and mixed-signal blocks would strengthen the framework and is a valuable future direction.
>
> 4. We appreciate the reviewer’s suggestion to improve accessibility. To support broader adoption and reproducibility, we have already released pretrained weights for the general forward GNN model (FALCON-main/checkpoints/best_gnn_model.pt), along with the fine-tuned GNN model on RVCO and the MLP topology classifier, all included in the supplementary material. All models are compact (e.g., 5.6MB for the GNN) and readily usable for fine-tuning or evaluation.
>
> Additionally, our released dataset includes a structured directory of circuit graphs and parameter-performance CSVs, and users can easily work with a smaller subset (e.g., 100k samples or a single topology family) without any changes to our codebase. We believe this enables efficient experimentation even for groups with limited resources, and we welcome further feedback from the community on additional lightweight configurations.
>
> **Limitations:**
>
> We thank the reviewer for this helpful suggestion. As noted, we already include a “Limitations” section at the end of the appendix in the supplementary material, summarizing key constraints of the current framework. We recognize that placing it in the supplement may have made it less visible. If the paper is accepted, we will move this section to the extra page of the main paper, and we will incorporate the additional limitations noted by the reviewers. We agree that making these boundaries explicit will help readers better understand the scope and applicability of FALCON.

---

> > ### Comment · Reviewer_RMej · 2025-08-05
> >
> > The provided extra evidence helps assess the maturity of the work. The multi-seed runs demonstrate that the good classification accuracy and single-digit prediction error are not due to fortunate initialization, and the choice to release lightweight checkpoints along with a scriptable 100 k-sample subset should make the pipeline viable for groups lacking a large RF simulation farm. Both aspects reinforce the reproducibility narrative.
> >
> > The explanation of why frequency can be either an input target or an output variable makes sense, even though it is still a design sketch at this point. When the framework is ultimately applied to a different frequency regime or mixed-signal block, its broader applicability to EDA will be much more evident.
> >
> > In general, the rebuttal confirmed that the primary results are sound and that the paper will be a valuable reference for end-to-end analog synthesis. For those reasons, I will retain my initial recommendation.

---

> > > ### Author Response · Authors · 2025-08-05
> > >
> > > Thank you very much for your thoughtful and constructive feedback. We truly appreciate your recognition of the “maturity of the work” and your insightful remarks on reproducibility and broader applicability. We're especially grateful for your view that the framework could become a “valuable reference” for future end-to-end analog synthesis efforts. Your comments are very encouraging, and we will reflect them carefully in the final version.

---

### Note · Authors · 2025-08-16

We sincerely thank all reviewers and the AC for their time and constructive feedback. Their input has been invaluable in clarifying our contributions and strengthening the presentation of this work.

In our rebuttal, we provided detailed clarifications along with additional experimental evidence. Below, we highlight several key points from those responses that directly addressed reviewer concerns:

- **Statistical robustness**: Reported results across multiple seeds for Stage 1 (MLP) and Stage 2 (GNN), presenting mean ± 95% CI, and including an ablation on initialization noise and variance for Stage 3 (gradient reasoning).
- **Flexible loss formulations**: Implemented two new performance losses: (1) a weighted loss to prioritize specific metrics, and (2) a threshold-based loss targeting minimum or maximum acceptable levels. The modular pipeline design supports adding new performance and layout objectives **with minimal effort and no retraining**, enabling future integration of user-defined costs.
- **Frequency generality**: Demonstrated how the framework can treat frequency as either a user-specified input (concatenated to the performance vector) or as an optimized output from gradient reasoning, enabling operation across diverse frequency ranges given appropriately diverse training data.
- **Applicability to other circuit domains**: Outlined extension to lower-frequency analog and mixed-signal blocks, noting that several studied topologies (e.g., voltage amplifiers) already map directly to low-frequency analog counterparts.
- **Limitations**: Moved the limitations section from the appendix into the main paper for greater visibility, and incorporated reviewer-suggested points to provide a more complete discussion.

We appreciate that the reviewers recognized the maturity of the work, its reproducibility, and its potential to serve as a valuable reference for future end-to-end analog circuit design research. We also note that all critical weaknesses were carefully considered and substantively discussed in the rebuttal, and no additional concerns were raised during the discussion.

In closing, we believe **FALCON** introduces a **first-of-its-kind**, **layout-aware**, **multi-stage** ML framework for analog circuit design, supported by a **large, high-fidelity dataset** and a **practical methodology**. We are committed to incorporating all clarifications and enhancements from the review process into the camera-ready version.

---

### Decision · Program_Chairs · 2025-09-17

**Decision:**

Accept (poster)

**Comment:**

This paper presents FALCON, a novel and comprehensive machine learning framework for fully automated, layout-aware analog circuit design, addressing the long-standing challenge of end-to-end specification-to-design synthesis.

The reviewers generally agree that the paper is well-written, technically sound, and makes a meaningful advance in ML for electronic design automation, with strong empirical results including >99% topology accuracy and sub-second design times.

While concerns were raised about the reliance on a fixed topology library, the simplicity of the current layout cost model, limited evaluation on non-mm-wave circuits, and initial lack of statistical robustness reporting, the authors provided thorough rebuttals with additional experiments and clarifications, addressing most issues satisfactorily.

Finally, all reviewers have provided borderline accept or accept ratings. This is a clear acceptance.